# The ClpXP protease and the ClpX unfoldase control virulence, cell division, and autolysis in *Streptococcus pneumoniae*

Viktor H. Mebus,[1,2] Supradipta De,[3] Larissa M. Busch,[4] Manuela Gesell Salazar,[4] Rabea Schlüter,[5] Uwe Völker,[4] Sven Hammerschmidt,[3] Dorte Frees[1]

**ABSTRACT**  In all living cells, molecular chaperones and ATP-dependent proteases are essential for protein homeostasis. The ClpXP protease is a chaperone-protease complex conserved between bacteria and mitochondria. Proteolytic activity resides in the caseinolytic protease (ClpP) subunits that associate to form a tetradecameric serine protease that degrades substrates recognized and unfolded by the ClpX ATPase. In the important human pathogen *Streptococcus pneumoniae* (*Spn*), ClpX but not ClpP was proposed to be essential for viability. We here follow up on this finding by characterizing phenotypes associated with depriving *S. pneumoniae* D39V of ClpX unfoldase or ClpXP activity. Virulence was examined in the *Galleria melonella* infection model and was found to be severely attenuated for both mutants, suggesting that ClpXP is essential for virulence. Inactivation of ClpXP also resulted in elevated extracellular levels of LytA and early onset of LytA-dependent autolysis, aberrant localization of teichoic acids, and diminished cell size, while depletion of ClpX resulted in defective division septa, heterogeneous cell sizes, elongated cell chains, and cell lysis. Accordingly, proteomic analysis revealed that ClpXP directly or indirectly controls proteins involved in teichoic acid synthesis and cell division and elongation. In summary, we show that ClpX alone and in complex with ClpP control critical processes such as virulence and cell division.

**IMPORTANCE** The human nose is colonized by the opportunistic pathogenic bacterium *Spn*, a leading cause of community-acquired pneumonia and a common cause of meningitis and sepsis. In *Spn*, the highly conserved ClpP protease is essential for virulence, and we here show that ClpP controls virulence through associating with the ClpX unfoldase. The essentiality of *Spn* ClpX has hampered its characterization. By constructing *Spn* mutants expressing a variant of ClpX that cannot interact with ClpP, we further show that the ClpXP protease controls vital processes such as cell size and autolytic activity. Depletion of ClpX resulted in even more severe phenotypes, suggesting that ClpX unfoldase activity independently of ClpP plays a more fundamental role in coordinating cell division and cell elongation. In summary, the presented work adds to our understanding of how a highly conserved chaperone-protease complex contributes to the growth and pathogenicity of a prominent bacterial pathogen.

**KEYWORDS** *Streptococcus pneumoniae*, ClpXP, chaperones, LytA, cell division, lipoteichoic acid, cell wall, *Galleria melonella*

All living cells rely on ATP-dependent chaperones and proteases for facilitating correct protein folding and for disposing cells of misfolded, unnecessary, or regulatory proteins (1, 2). Bacteria and mitochondria of eukaryotic cells share the caseinolytic protease (ClpP), a self-compartmentalizing serine protease composed of 14 subunits that sequester the active serine residues within an inaccessible hollow barrel (3). Substrates are recognized, unfolded, and threaded into the proteolytic chamber

Address correspondence to Dorte Frees, df@sund.ku.dk, or Viktor H. Mebus, vhm@sund.ku.dk.

The authors declare no conflict of interest.

See the funding table on p. 19.

by cognate Clp ATPases belonging to the protein superfamily of ATPases associated with diverse cellular activities (1, 4, 5). Bacterial ClpP has the ability to associate with multiple Clp ATPases (ClpA, ClpC, ClpE, and ClpX); however, only ClpX is conserved among mitochondria and gram-positive and gram-negative bacteria (6, 7). In mitochondria, the ClpX unfoldase activity independently of ClpP contributes to vital processes such as heme biosynthesis and nucleoid distribution (8, 9). The ClpP-independent roles of ClpX in bacterial cell biology remain relatively unexplored, but we recently showed that ClpX has an essential role in bacterial cell division that does not involve ClpP (10, 11). Additionally, earlier studies demonstrated that ClpX-mediated unfolding is important for the function of the phage MuA transposase independently of ClpXP-mediated degradation (12). The current study was conducted to determine the role of the ClpXP protease and the ClpP-independent role of ClpX in the important pathogen, *Streptococcus pneumoniae*.

The gram-positive bacterium, *S. pneumoniae* or the pneumococcus, is a common colonizer of the mucosal surface of the upper respiratory tract of healthy humans and the leading cause of community-acquired pneumonia worldwide (13). In children, the elderly, and immunocompromised individuals, pneumococcus is also associated with life-threatening, invasive infections such as sepsis and meningitis, and globally, pneumococci are estimated to be causing ~800,000–1,000,000 deaths annually (14, 15). Infections are preferentially treated with antibiotics derived from penicillin, the so-called β-lactams that today are still the most prescribed class of antibiotics globally (16). Penicillin and related antibiotics bind to and inactivate enzymes that catalyze the cross-linking of peptidoglycan in the bacterial cell wall (17–19). Pioneering studies with *S. pneumoniae* showed that bacterial killing involves cell lysis elicited by the unsynchronized activation of cell wall hydrolases, the so-called autolysins (20, 21). In *S. pneumoniae*, the N-acetylmuramoyl L-alanine amidase LytA is the major autolysin responsible for penicillin-induced bacteriolysis (21, 22). During exponential growth conditions, pneumococci are normally not susceptible to LytA-mediated lysis but become sensitive in stationary phase (23). This switch in sensitivity has been suggested to be linked to a growth phase-dependent synthesis of teichoic acids (TAs) that are anionic glycopolymers, which are either membrane-linked lipoteichoic acids (LTAs) or peptidoglycan-linked wall teichoic acids (WTAs) (24, 25). TAs are signature molecules of the cell envelope of gram-positive bacteria, where they can make up to 60% of the cell wall biomass and play critical roles in colonization, virulence, cell division, and osmotic stress tolerance (26–30). In contrast to most gram-positive bacteria, pneumococcal WTA and LTA chains have identical and unique repeating unit structures that are decorated by phosphorylcholine residues, which serve as anchors for the choline-binding proteins that include peptidoglycan hydrolases like LytA and major virulence factors such as CbpG (cleaves host extracellular matrix [31]), pneumococcal surface protein A (PspA, inhibits macrophage opsonization and complement activation [32–35]), and pneumococcal surface protein C (PspC, involved in adherence and complement inactivation [36–40]).

The pathogenicity of bacteria is critically dependent on their ability to survive in a host, and as elegantly summarized in a recent review, the ClpXP protease controls many different virulence-related pathways, including control of stress responses, biofilm formation, and virulence effector protein production (41).

Nonetheless, mutations in the *clpP* and *clpX* genes have on multiple occasions been identified in clinical *Staphylococcus aureus* strains isolated from patients, despite the fact that the ClpXP protease is essential for virulence in animal models of *S. aureus* infections (42–48). This paradoxical finding may be linked to an unexpected role of ClpXP in evading the immune system and determining *S. aureus* susceptibility to important classes of antibiotics targeting the cell wall, including β-lactams, daptomycin, or vancomycin (49–54). The underlying mechanisms remain unexplained. However, genetic characterization of suppressor mutations that alleviate the growth defect of *S. aureus clpX* mutants localizes in the widely conserved cell division gene, *ftsA*, or in the LTA biosynthesis gene, *ltaS*, revealing a functional link between ClpX(P) and cell wall

synthesis in this important pathogen (11, 55). To investigate whether this function of ClpX(P) is conserved in other firmicutes, we here assess the role of the ClpX unfoldase and ClpXP in virulence, cell division, and autolysis in pneumococcus.

## RESULTS

### Depletion of ClpX is selective for mutations in the SPV_1595 gene

Clp ATPases interact with ClpP via an exposed IGF loop that docks into clefts on the surface of the ClpP multimer, and substituting the isoleucine residue in the IGF motif is known to abolish ClpXP proteolysis (56, 57). To study the function of the ClpXP protease in *S. pneumoniae* D39V, we therefore utilized the Janus cassette to introduce an I267E substitution in the IGF tripeptide of ClpX (58). The resulting strain, D39V*clpX*$_{I267E}$, retains functional ClpP and ClpX ATPase activity, hence allowing us to specifically study phenotypic changes associated with inhibiting ClpXP protease activity (Fig. 1A and B). In agreement with ClpX being essential in D39V, we were unsuccessful in deleting the *clpX* gene using standard Janus cassette cloning. To study phenotypic changes associated with depletion of ClpX, we instead transcriptionally fused the *clpX* gene to the zinc-inducible *czcD* gene (59). This enabled us to delete the *clpX* gene from its native genomic location in D39V*czcD::clpX* cells grown in the presence of zinc to induce ectopic ClpX expression (Fig. 1B). After 4 h of growth in the absence of Zn, ClpX was no longer detectable in protein extracts from the D39VΔ*clpX czcD::clpX* strain, demonstrating efficient depletion of ClpX (Fig. 1C). Despite the fact that the *clpX* gene was deleted in cells expressing ClpX, whole genome sequencing revealed that the Δ*clpX czcD::clpX* strain had acquired a premature stop codon in the SPV_1595 gene (position S160*), whose inactivation was previously shown to suppress lethality associated with deletion of the *clpX* gene in the D39V-derived strains, R6 and Rx (60). In contrast, the introduction of the *clpX*$_{I267E}$ allele was not associated with additional genetic changes. Therefore, our data support that the gene product of the SPV_1595 gene, which together with the neighboring SPV_1594 gene has been proposed to encode a toxin/antitoxin addiction module, imposes a lethal phenotype in D39V lacking ClpX unfoldase activity but not in cells lacking only ClpXP protease activity. Despite the premature stop codon in the SPV_1595 gene, phenotypes associated with ClpX depletion can be deduced by comparing D39V Δ*clpX czcD::clpX* grown in in the absence of Zn²⁺ (ClpX depleted) or in the presence of Zn²⁺ (ClpX ectopically expressed).

### ClpXP inactivation accelerates LytA-mediated autolysis, while ClpX depletion results in a hyper-chained phenotype

Pneumococcus is well known for its ability to develop genetic competence under specific conditions, and previous studies indicate that ClpXP contributes to the inhibition of competence development under non-permissive conditions (61, 62). We therefore transcriptionally fused the firefly luciferase to the native *ssbB* locus (*ssbB-luc*) to follow growth and competence development in our strains during growth conditions that are either permissive (C + Y at pH 7.8) or non-permissive (C + Y at pH 7.0–7.3) for competence development (Fig. S1A and B) (63). At both pH values, the growth of the mutants was identical to the growth of the D39V wild type (WT) (Fig. 1D; Fig. S1C; exponential doubling time at pH 7.8: WT = 35.1 ± 3.6 min, D39V *clpX*$_{I267E}$ = 35.9 ± 2.8 min, D39VΔ*clpX czcD::clpX* = 35.8 ± 4.2 min). As expected, WT cells only expressed luciferase under conditions that are permissive for competence, and this was also true for both of the two mutants that nevertheless showed diminished luciferase expression at competence permissive conditions (pH 7.8, Fig. S1A and B). Therefore, our data do not support that ClpXP represses competence development. Instead, we observed that the ClpX$_{I267E}$ variant accelerated the onset of autolysis in C + Y media (pH 7.8, Fig. 1D, and pH 7.0, Fig. S1C), a phenotype that was also observed in the Δ*clpX czcD::clpX* mutant, but only if cells were grown in the absence of ZnCl₂ (Fig. 1D; Fig. S1D). Stationary-phase autolysis of *S. pneumoniae* is associated with the choline-binding LytA cell wall hydrolase (20, 21, 23). In

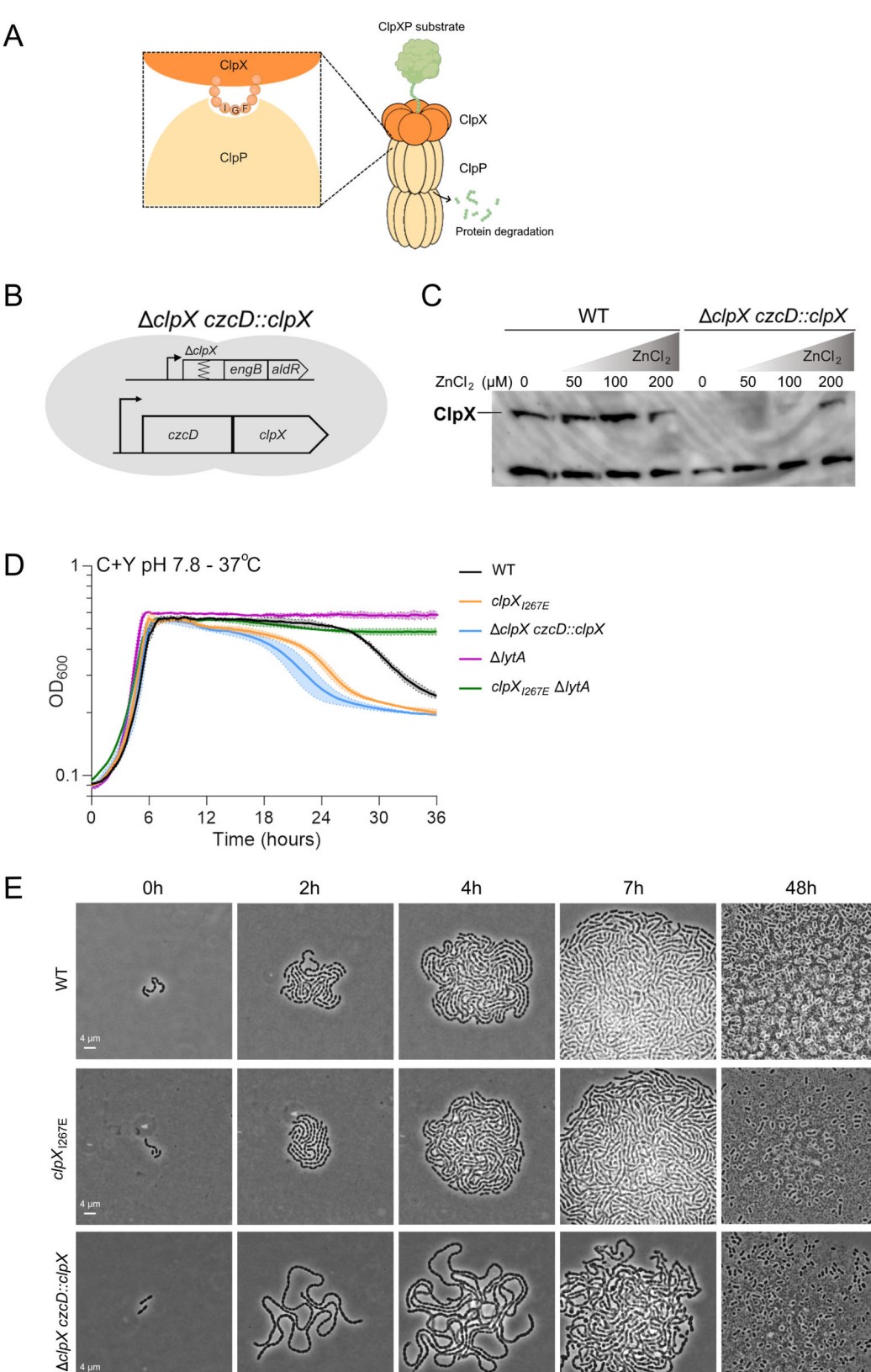

**FIG 1** Comparative analysis of mutants to study growth phenotypes associated with inactivation of the ClpXP protease or depletion of ClpX in *S. pneumoniae*. (A) Schematic drawing showing that ClpX subunits interact with ClpP proteolytic subunits via the exposed IGF tripeptide loop. (B) To allow for conditional depletion of ClpX, the native *clpX* gene, localized upstream (Continued on next page)

**Fig 1 (Continued)**

of the essential *engB* gene, was deleted in a strain having an extra copy of the clpX gene transcriptionally fused to the zinc-inducible gene, czcD, thereby allowing Zn-dependent expression of ClpX as shown by ClpX immunoblotting (C): here, the D39V wild-type (WT) and D39V Δ*clpX czcD::clpX* cells were grown to an $OD_{600}$ of ~0.4 in the absence or presence of $ZnCl_2$ (50, 100, or 200 µM). The lower band protein is an unknown unspecific target of the ClpX antibody, here used as a loading control. (D) Growth of WT, *clpX* mutants, and *lytA* mutants in C + Y medium, pH 7.8, at 37°C for 36 h. (E) Extended simultaneous time-lapse microscopy with WT, *clpX_{I267E}*, and Δ*clpX czcD::clpX* strains grown on C + Y and 1% agarose at 37°C at pH 7.8.

agreement with this notion, lysis was abrogated when the *lytA* gene was deleted, showing that LytA is causing the accelerated lysis of cells devoid of ClpX or ClpXP (Fig. 1D; Fig. S1C).

Growth was additionally investigated by following the growth of single cells on C + Y + 1% agarose at two different pH values (pH 7.0 or 7.8) by extended time-lapse microscopy (Fig. 1E; Fig. S2). The images revealed that both the D39V WT and the mutant strains grow in chains of extended length when grown at pH 7.8 (Fig. 1E) instead of pH 7.0 (Fig. S2). In particular, ClpX-depleted cells grow in very long chains at pH 7.8 (Fig. 1E). The most prominent phenotypic difference between WT and cells lacking ClpXP protease activity was the accelerated autolysis phenotype that was observed after 40 h as compared to 48 h for the WT. Collectively, the results suggest that ClpX, independently of the ClpXP protease, modulates chain separation, whereas inactivation of the ClpXP protease accelerates the onset of autolysis.

## ClpXP is essential for virulence in the *Galleria mellonella* infection model

Next, we tested the virulence of the two *clpX* mutant strains in the *Galleria mellonella* larvae model of infection (27). The larvae have an immune system analogous to the human innate immune system but lack adaptive immunity, making this model suitable to study early host-pathogen interactions (64). The virulence of each strain was determined in three biological replicates by infecting 10 larvae with approximately 3 × 10⁵ bacteria injected into the rearmost pair of legs (as a negative control, 10 larvae were infected with 0.9% NaCl). Prior to infection, the D39VΔ*clpX czcD::clpX* strain was grown with or without 200 µM $ZnCl_2$ to allow for determining the role of ClpX in virulence. The relative virulence of the strains was assessed by counting the relative number of white, live larvae and the number of darkened, dead larvae on a daily basis for a 7-day period (see representative images in Fig. 2) (65). Interestingly, the lethality of D39V was significantly diminished either by depleting ClpX or by abolishing ClpXP activity (Fig. 2). The association of reduced virulence with ClpX depletion is supported by the observation that the D39V Δ*clpX czcD::clpX* strain partly regained lethality when pre-grown in the presence of $ZnCl_2$ to allow for ectopic ClpX expression at the time of infection ($T = 0$). Together, the results suggest that ClpX contributes to *S. pneumoniae* lethality via a ClpXP-dependent pathway (Fig. 2).

## ClpXP inactivation increases extracellular LytA levels

To investigate whether the accelerated lysis of D39V lacking ClpXP activity is associated with changes in LytA protein abundance, LytA immunoblotting was performed with cellular and extracellular protein extracts derived from exponential ($OD_{600} = 0.3$) or early stationary cultures ($OD_{600} = 0.9$) of the WT and *clpX* mutant strains. Interestingly, the extracellular LytA signal was 68- and 49-fold higher in medium derived from exponential cultures of D39V*clpX_{I267E}* or D39V depleted for ClpX compared to the medium derived from D39V WT (Fig. 3A), whereas the levels of cellular LytA did not change significantly between the three strains (Fig. 3A). LytA does not contain a signal peptide, and translocation of LytA to the extracellular space is believed to be mediated by cellular lysis (23, 66). We tested for cell lysis by probing with antibodies specific for enolase (cytoplasmic) and pneumolysin, which, similar to LytA, is exported out of the cell despite lacking a recognizable signal peptide. However, we found that levels of both proteins are

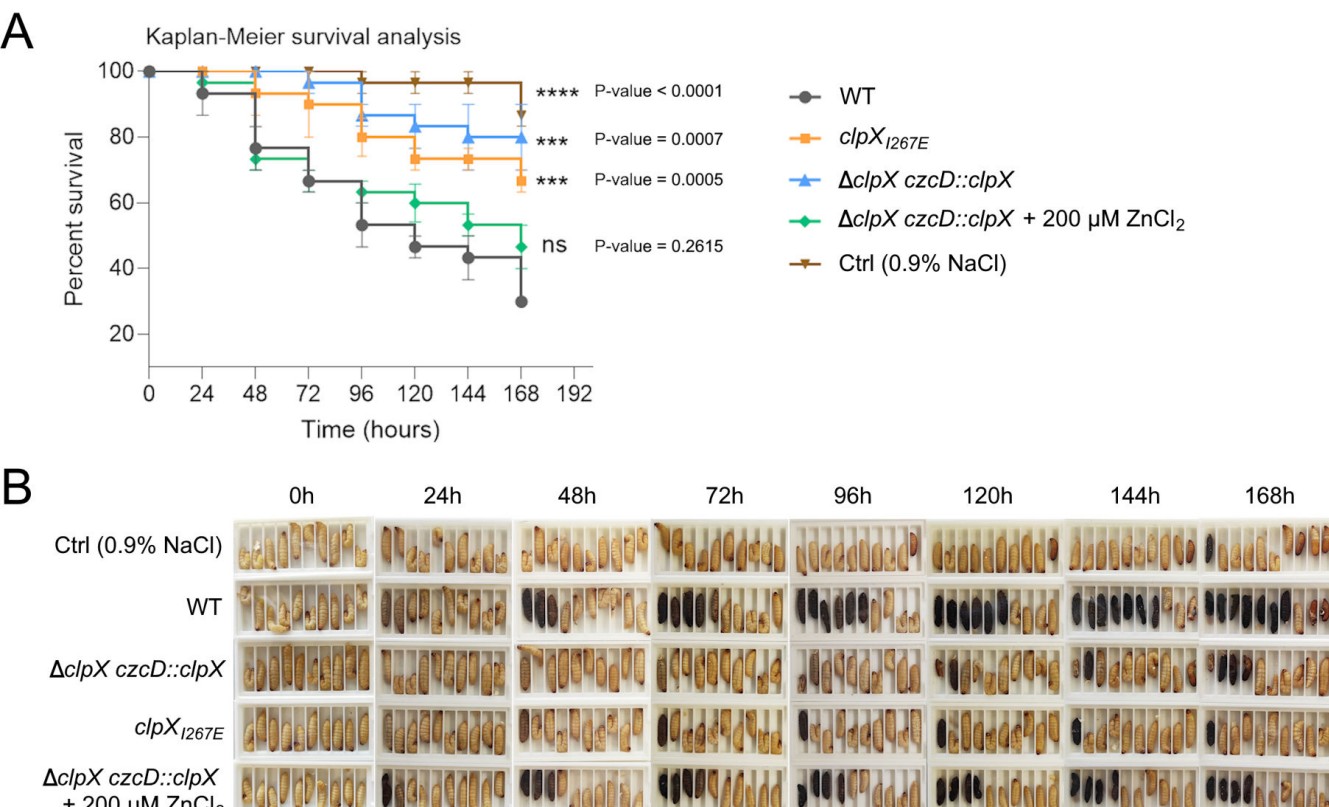

**FIG 2** The ClpXP protease is important for virulence in *Galleria melonella*. (A) Kaplan-Meier diagram of *G. melonella* survival post-infection with WT and *clpX* mutants, observed for 7 days. Survival curves are plotted as means of three separate infection assays with standard errors of the mean. ClpX levels were restored by adding 200 µM ZnCl$_2$ to the growth medium of Δ*clpX czcD::clpX*. P values were calculated, comparing each condition to the wild type, with a log-rank (Mantel-Cox) test. Asterisks represent P values, where $P < 0.001$ (***), $P < 0.0001$ (****), and $P > 0.05$ (ns). (B) Sample image of one of the three experiments injected with their respective strains. Deaths were accounted for every 24 h for 7 days, where blackened larvae indicate deceased larvae.

unaltered in the mutants ruling out increased lysis (Fig. 3A). Furthermore, quantitative reverse transcription PCR (RT-qPCR) revealed that the increase in LytA levels is not paralleled by increased *lytA* mRNA levels (Fig. 3B). We conclude that the accelerated autolysis observed for D39V devoid of ClpXP activity correlates with greatly enhanced levels of extracellular LytA, despite the fact that transcription of the *lytA* gene and cellular levels of LytA are not affected in cells lacking ClpXP proteolytic activity. For now, the mechanisms underlying the increase in extracellular LytA levels remain obscure.

In *S. pneumoniae*, unsynchronized activation of LytA is responsible for penicillin-induced bacteriolysis (21, 22), and we wondered if the accelerated buildup of LytA in the growth medium would make D39V*clpX*$_{I267E}$ and D39V depleted for ClpX more susceptible to penicillin killing. However, this turned out not to be the case (Fig. 3C).

### Inactivation of ClpXP results in aberrant septal TA labeling and reduces cell size, while complete depletion of ClpX results in heterogeneous cell size

Extracellular LytA attaches to the pneumococcus cell wall through binding non-covalently to the phosphorylcholine units on TAs, and TAs are believed to play a key role in controlling LytA activity (24, 25). Hence, we asked whether changes in TA content or deposition could explain LytA accumulation extracellularly in exponential cultures of our two *clpX* mutants. Growth of *S. pneumoniae* depends on exogenously supplied choline, and because choline is exclusively integrated into TA, labeling of cellular TA can be achieved by growing cells in the presence of 1-azidoethyl-choline prior to imaging and utilizing the sDIBO Alexa Fluor-488 for a click-chemistry reaction as described by reference 67 (Fig. 4A). Phase-contrast images first revealed that inactivation of ClpXP

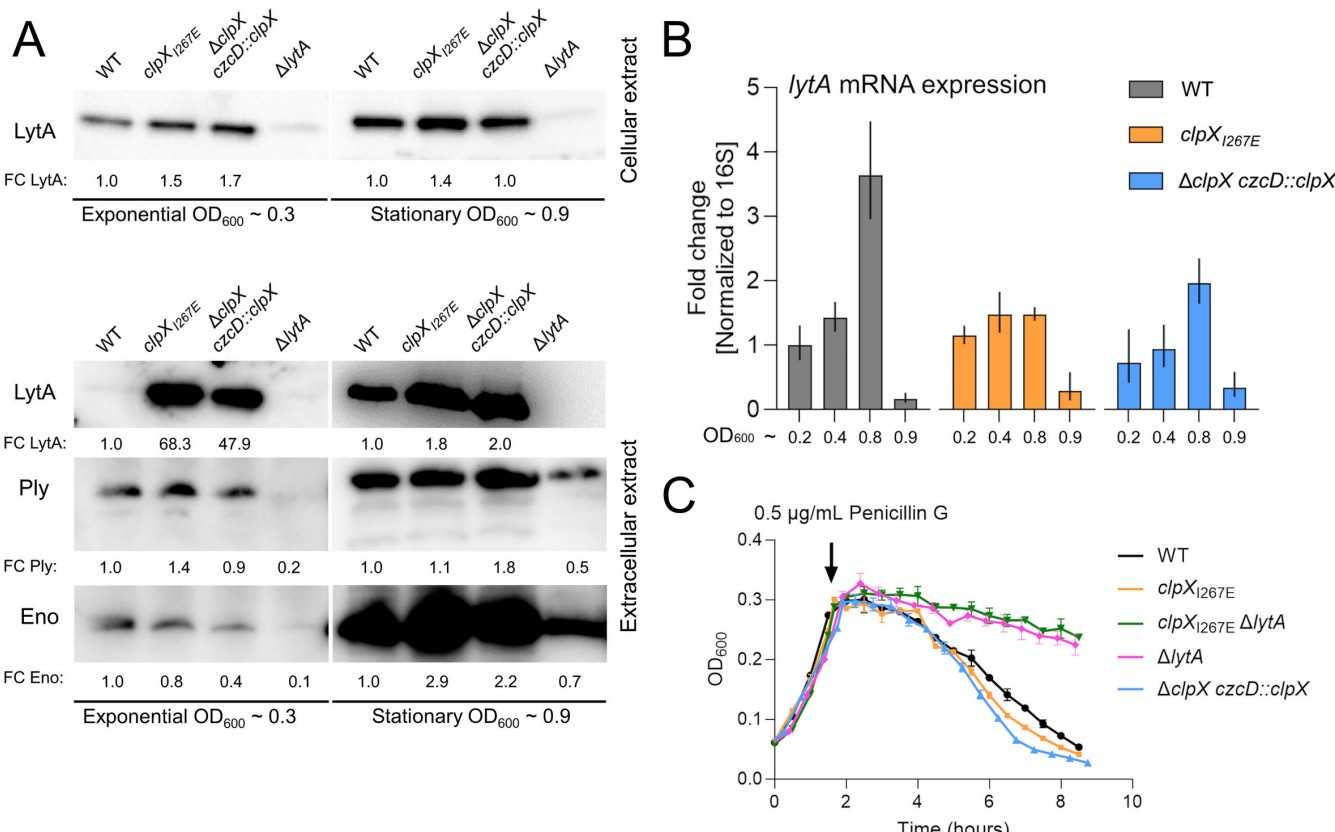

**FIG 3** ClpXP inactivation increases extracellular LytA levels but does not increase penicillin killing. (A) LytA immunoblotting performed using cellular extracts and corresponding supernatant extracts from WT, $clpX_{I267E}$, $\Delta clpX$ $czcD::clpX$, and $\Delta lytA$ strains grown in 40 mL C + Y media at pH 7.8 (exponential cultures, $OD_{600}$ ~0.3: $clpX_{I267E}$: $P$ value = 0.058, $\Delta clpX$ $czcD::clpX$: $P$ value = 0.224; stationary-phase cultures, $OD_{600}$ ~0.9: $clpX_{I267E}$: $P$ value = 0.054, $\Delta clpX$ $czcD::clpX$: $P$ value = 0.308). (B) $lytA$ mRNA expression relative to 16S was quantified with RT-qPCR using RNA samples extracted from WT, $clpX_{I267E}$, and $\Delta clpX$ $czcD::clpX$ $OD_{600}$ cells grown in C + Y medium to $OD_{600}$ ~0.2, 0.4, and 0.8 and after 2 h at $OD_{600}$ ~0.9 (stationary phase). (C) Strains of WT, $clpX_{I267E}$, $\Delta clpX$ $czcD::clpX$, $\Delta lytA$, and $clpX_{I267E}$ $\Delta lytA$ were grown in C + Y at pH 7.8 until $OD_{600}$ ~0.3 before the addition of 0.5 µg/mL penicillin G. $OD_{600}$ was measured every 30 min.

protease activity is associated with a slight but significant reduction in cell size (WT [$n$ = 264], mean size = 1.019 ± 0.28 um², as compared to $clpX_{I267E}$ [$n$ = 241], mean size = 0.9633 ± 0.29 µm²; $P$ value = 0.028) – Fig. 4B. Depletion of ClpX, on the other hand, resulted in heterogeneously sized cells ($\Delta clpX$ $czcD::clpX$, mean size = 1.116 ± .54, $P$ value = 0.031, $n$ = 196), an effect that was reversed by inducing ClpX expression by growing cells in the presence of $ZnCl_2$ (Fig. 4B). To study the cellular TA distribution in cells in different stages of the cell cycle, the fluorescence intensity of the TA signal was measured in 1,000 cells arranged according to their length (Fig. 4C), and for cells belonging to each of the depicted stages (I, II, III, and IV), heatmaps were built from the normalized TA signal (Fig. 4C and D). This analysis revealed that the cell wall of cells in the first phase of the cell cycle (phase I = non-septating cells) displays a uniform TA signal in the outer wall (Fig. 4C and D). In cells in phase II or III, the TA signal is most intense at the cell poles, and in all strains, an unlabeled gap becomes apparent at mid-cell (Fig. 4A, C, and D). Interestingly, the mid-cell gap devoid of TA labeling is much wider in cells either depleted for ClpX (phases II and III) or expressing the $ClpX_{I267E}$ variant (phases III and IV) (Fig. 4C and D). Of note, the first cell in a chain of cells generally has very diminished TA labeling in all three strains (Fig. 4A). Therefore, inactivation of ClpXP proteolytic activity is associated with reduced cell size and aberrant TA distribution in septating cells, while depletion of ClpX is associated with heterogeneously sized cells and aberrant TA localization.

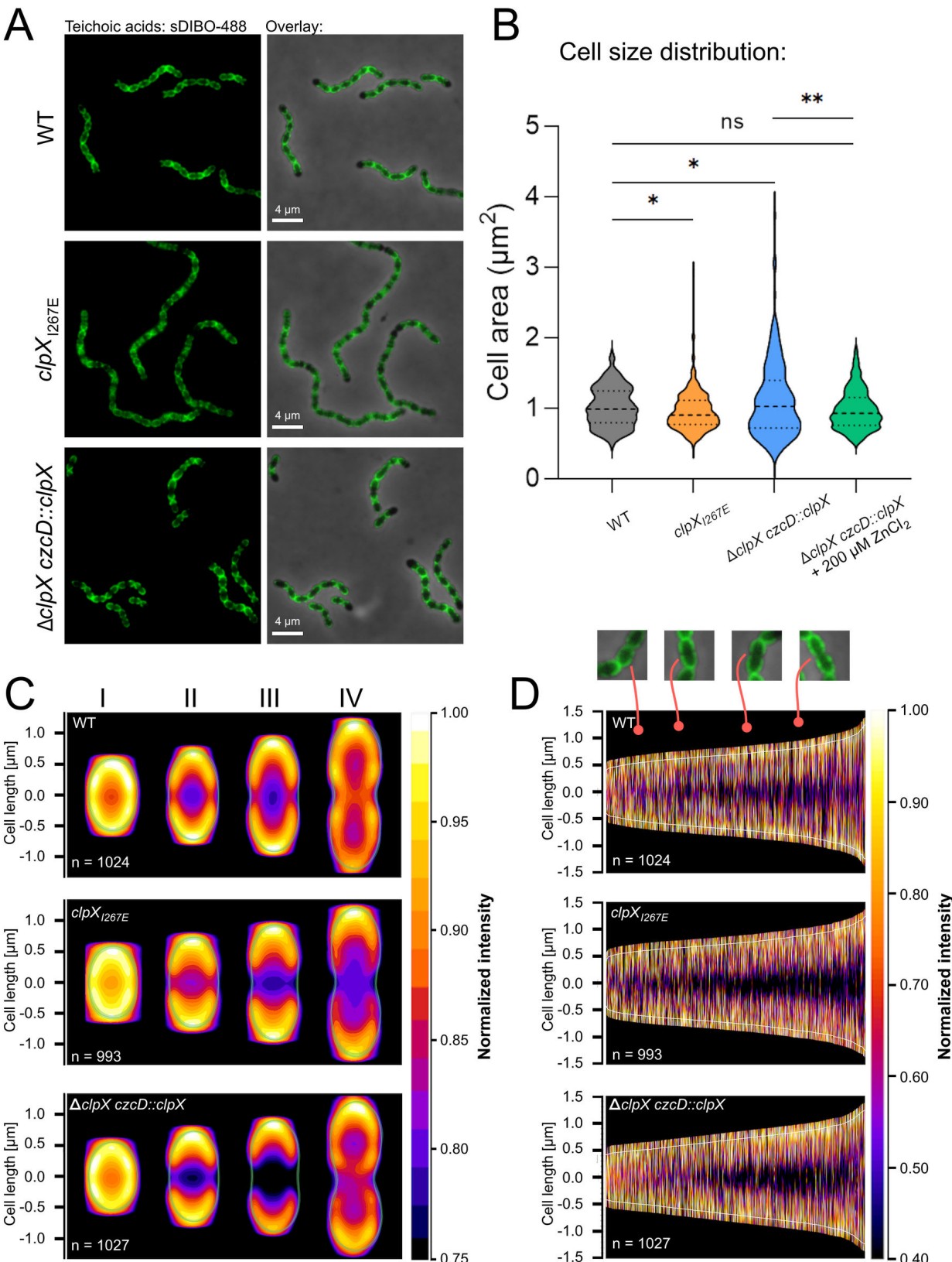

**FIG 4** Inactivation of ClpXP protease activity alters the TA staining patterns during the late division state. (A) Phase-contrast and fluorescent images of teichoic acids labeled with 5 µg/mL 1-azidoethyl-choline and 50 µM sDIBO Alexa Fluor-488 for WT, *clpX_{I267E}*, and Δ*clpX czcD::clpX* grown until OD_{600} ~0.3. (B) Cell sizes quantified in Fiji, using phase-contrast images of three biological replicates of WT (*n* = 264), *clpX_{I267E}* (*n* = 241, *P* value = 0.028), Δ*clpX czcD::clpX* (*n* = 196, *P* value

Fig 4 (Continued)

= 0.031), and Δ*clpX czcD::clpX* + 200 μM ZnCl$_2$ ($n$ = 224, $P$ value = 0.13). Asterisks represent $P$ values from a two-tailed Welch's t-test where $P < 0.05$ (*), $P < 0.01$ (**) and $P > 0.05$ (ns). Δ*clpX czcD::clpX* was additionally compared to Δ*clpX czcD::clpX* + 200 μM ZnCl$_2$ ($n$ = 224) using a two-tailed Welch t-test, revealing a $P$ value of 0.003. (C) Heatmaps built from the normalized TA signal, for each strain (WT, $n$ = 1,024; *clpX*$_{I267E}$, $n$ = 993; and Δ*clpX czcD::clpX*, $n$ = 1,027) subcategorized into cell shapes, corresponding to the four main stages of the cell cycle using the MicrobeJ plugin for Fiji. (D) Demographs showing normalized TA signal for each strain, sorted according to cell length using the MicrobeJ plugin for Fiji.

## ClpX chaperone depletion results in irregular cell morphology, septal abnormalities, and pili-like structures

To further investigate the role of ClpX and ClpXP in pneumococcus cell division, cells were additionally imaged with transmission electron microscopy (TEM) (Fig. 5A) and scanning electron microscopy (SEM) (Fig. 5B). Strikingly, both TEM and SEM images confirmed that depletion of ClpX resulted in more severe morphological defects than inhibiting ClpXP activity. In TEM images, some cells depleted for ClpX displayed septal irregularities and cell lysis at the septal site where a characteristic thinning of capsule was observed (Fig. 5A). In SEM images, cells depleted of ClpX were characterized by the presence of fimbria-like structures protruding from the cell poles and mid-cell (Fig. 5B); the phenotypes were reversed by growing cells in the presence of 200 μM ZnCl$_2$, showing that the phenotypes are associated with the depletion of ClpX (Fig. S3).

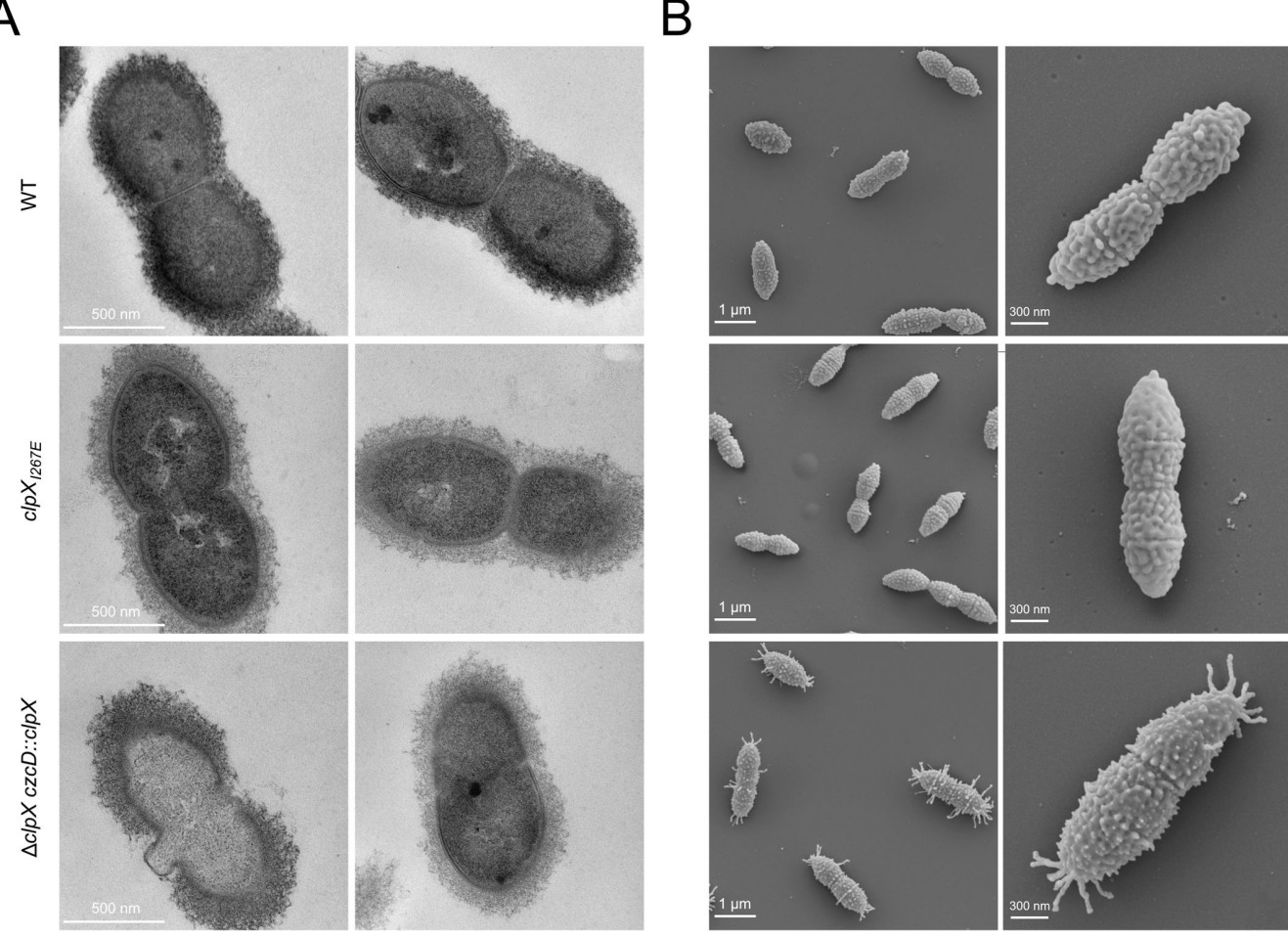

**FIG 5** ClpX depletion results in septal abnormalities. (A) The indicated strains were grown in C + Y pH 7.8 media at 37°C until mid-log phase before fixation and TEM imaging. (B) SEM images were acquired following growth (OD ~0.4) in RPMI-based minimal medium with 0.25% yeast extract and fixed with 2.5% glutaraldehyde. TEM and SEM imaging were done twice.

Taken together, the images indicate that ClpX unfoldase activity independently of ClpP contributes to pneumococcal cell division.

## Proteomic changes in pneumococcus devoid of ClpXP protease activity

As a first attempt to delineate the pathways by which inactivation of ClpXP regulates phenotypes such as decreased virulence and altered cell morphology, we determined the changes in the cellular pneumococcal proteome caused by expression of the ClpX$_{I267E}$ variant. Total cellular protein samples were extracted from cells harvested in exponential phase (OD$_{600}$ ~0.5) (Fig. 6) and in post-exponential stationary phase (OD$_{600}$ ~1.5) (Fig. S4B). To avoid the heterogeneity associated with the induction of genetic competence, a process that affects transcription of 5%–10% of the genome (68), the strains were cultured in RPMI medium where competence development is inhibited (Fig. S4A). In total, 1,226 proteins were identified with at least two peptides, and these proteome profiles were used to identify proteins differentially expressed between *S. pneumoniae* WT cells and cells producing the ClpX$_{I267E}$ variant. In line with the notion that pneumococci do not develop competence in RPMI medium, the CbpD muralytic

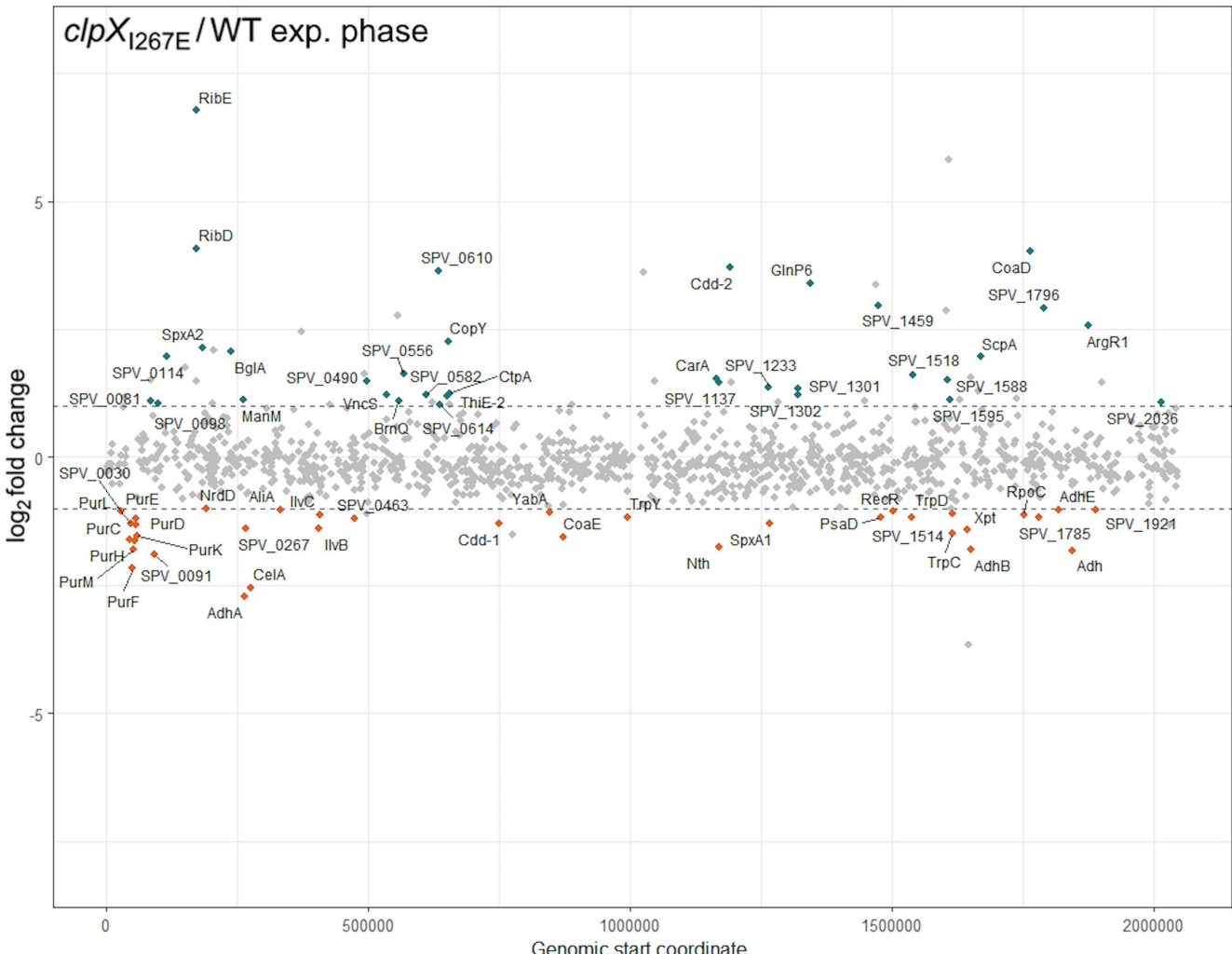

**FIG 6** Alterations in the proteome profile associated with inactivation of ClpXP. Proteins (≥2 identified peptides) were quantified for the D39V *clpX$_{I267E}$* mutant relative to the D39V WT. Each protein is plotted with a solid dot, according to the genomic start coordinate (*x*-axis), in relation to the relative log$_2$ transformed fold change (*y*-axis). Upregulated proteins (log$_2$ fold change >1) with an adjusted *P* value of <0.05 were in green-blue and labeled accordingly. Downregulated proteins (log$_2$ fold change <−1) with an adjusted *P* value of <0.05 were in orange and labeled accordingly. Non-significant and non-changing proteins were plotted as gray dots.

fratricin and the immunity protein ComM, known to be expressed only during competence, were among the proteins that could not be detected in the two strains (see extended data). The proteome profiling identified 34 proteins that displayed significantly ($Q$ value <0.05) higher protein levels (≥2-fold) in cells lacking ClpXP activity (Fig. 6; Table S1). Interestingly, the SPV_1595 protein that is associated with lethality of ClpX inactivation in D39V and derived strains (60) is almost threefold higher in cells devoid of ClpXP (Fig. 6; Table S1). Such increases in protein abundances may reflect the impact of lack of ClpXP protease on direct targets. However, the upregulation may also reflect that ClpXP indirectly impacts expression at the level of transcription or translation. Such indirect effects might also explain the lower abundance of 36 proteins in D39V$clpX_{I267E}$ compared to the wild type, including proteins involved in purine biosynthesis and four alcohol dehydrogenases (Adh, AdhA, AdhB, and AdhE) (Fig. 6; Table S1).

The highly conserved transcriptional regulator SpxA has been identified as a substrate of ClpXP across the *Firmicutes* phylum (69). Streptococcal species typically contain two SpxA paralogs, SpxA1 and SpxA2 (70). In support of the idea that SpxA2 is a ClpXP substrate also in the pneumococcus, the protein was found to be highly increased in abundance in the proteome of D39V $clpX_{I267E}$ compared to the wild type (4.5-fold/6.5-fold in exponential/stationary growth phase). In contrast, SpxA1 displayed even two- to threefold lower levels in the D39V $clpX_{I267E}$ strain, suggesting that SpxA1 is not degraded by ClpXP in pneumococci (Table S1). The other proteins displaying highly increased levels in exponential phase in D39V $clpX_{I267E}$ belonged to various classes.

## DISCUSSION

The ClpX unfoldase was considered to be essential in *S. pneumoniae* but not essential in other *Streptococcus* spp., such as *Streptococcus suis* (71), *Streptococcus mutans* (72), or *Streptococcus pyogenes* (73). In line with previous results, we here show that the essentiality of pneumococcal ClpX is linked to the SPV_1595 gene, which together with the neighboring SPV_1594 gene has been proposed to encode a horizontally acquired toxin/antitoxin addiction module (74). In contrast, D39V cells lacking ClpXP activity while retaining ClpX unfoldase activity are viable despite accumulating the SPV_1595-encoded protein. Together, these results emphasize that the lethal activity of SPV_1595 is counteracted by ClpX unfoldase activity via a mechanism not involving ClpP. This finding is consistent with previous studies showing that *S. pneumoniae* R6Δ*clpP* is viable, although that transcription of the SPV_1594–1595 operon is elevated 10-fold (62, 75).

Deletion of the *S. pneumoniae clpP* gene is known to cause pleiotropic phenotypes, including strongly attenuated virulence in murine lung and sepsis infection models; however, it has not been investigated if these phenotypes are linked to depriving cells of ClpCP, ClpEP, or ClpXP proteolytic activity (75, 76). Here, we indicate that ClpXP contributes to virulence by showing that pneumococcus carrying a ClpX variant that cannot associate with ClpP displays reduced lethality in the *G. mellonella* larvae model of infection (77). When we searched our proteomic data for changes that could explain the decrease in virulence, we did not observe reduced expression of classical virulence factors such as the CBPs. Also, the expression of pneumolysin was unchanged in the mutant. Release of pneumolysin is promoted by alcohol dehydrogenases, and inactivation of this class of enzymes is associated with decreased virulence (78). Strikingly, four out of five pneumococcal alcohol dehydrogenases are reduced in protein abundance in the D39V $clpX_{I267E}$ mutant, a finding that may at least contribute to the reduced lethality of the mutant.

In most bacteria, the ClpX chaperone is dispensable for growth; however, exceptions include *Caulobacter crescentus*, where the ClpXP protease is indispensable for cell cycle progression and chromosome replication (79); *Mycobacterium smegmatis*, where ClpX depletion results in non-viable filamentous cells (80); and *S. aureus*, where deletion of the *clpX* gene results in severe cell division problems and cell lysis at 30°C (10, 11). The common theme emerging from these studies is that the essentiality of ClpX or ClpXP is linked to a role in cell division. In support of the idea that ClpX is also

contributing to cell division-related processes in pneumococcal cells, we here show that cells depleted for ClpX are characterized by heterogeneous cell size, abnormal septum formation, diminished TA at the septal site, a buildup of extracellular LytA in exponential cultures, and accelerated LytA-dependent lysis. The latter two phenotypes seem to be associated with inactivation of the ClpXP protease. The heterogeneous cell size and the defective division septa associated with ClpX depletion support the idea that ClpX unfoldase activity, independently of ClpP, plays a more fundamental role in coordinating cell division and cell elongation. Septum synthesis and cell elongation are mediated by multiprotein complexes—the divisome and elongasome, respectively—whose assembly and activity are strictly regulated in space and time (81, 82). Accordingly, cells of heterogeneous sizes can be observed following many types of disturbances, including a knockdown of transcription of genes in translation, cell division, TA biosynthesis, and membrane synthesis (61). Hence, it will be far from trivial to elucidate how ClpX contributes to controlling cell size and coordinating cell division in *S. pneumoniae*. Of note, *S. aureus* ClpX seems to promote an early step of cell division as the septum synthesis defect was specifically alleviated by a spontaneously acquired mutation in the highly conserved cell division gene, *ftsA*, that accelerated septum initiation (11).

Inactivation of ClpXP alone resulted in shortened pneumococcal cells, indicative of impaired cell elongation. If ClpXP proteolysis is directly controlling the stability of proteins involved in TA synthesis, cell division, or cell elongation, we would expect such proteins to accumulate in D39V expressing the $ClpX_{I267E}$ variant. Our proteome profiling of exponentially growing cells, however, only detected a slight (1.5-fold) but significant accumulation of the cell division proteins SepF, PBP2a, and PBP3 and the cell elongation protein MreC, whereas FtsA, FtsZ, Pbp2b, DivIA, and GpsB were ≥1.5-fold lower (Table S1). Therefore, these proteins are rather indirectly controlled by ClpXP protease activity. In the stationary phase, three proteins with a proposed function in TA biosynthesis, cell elongation, or cell division were among the most upregulated proteins, namely, the lipoteichoic acid ligase, TacL (4.0-fold up, *P* value <0.5), the cell division protein MraZ (2.9-fold up, *P* value <0.2), and the RodA peptidoglycan polymerase associated with cell elongation (2.3-fold up, *P* value <0.1) (Fig. 6; Table S1). Additionally, the proteomics data set shows that the D39V *clpX_{I267E}* strain has increased levels of an enzyme that modifies the choline pattern on TA, namely, the phosphorylcholine esterase CbpE or Cpe that was 2.9-fold higher in the mutant; however, this change was not significant (*P* value = 0.5). Of these proteins, the teichoic acid ligase TacL is of special interest, as TacL via LTA anchoring was proposed to be critical in protecting pneumococcal cells from LytA-dependent lysis during exponential growth (25). According to this model, TacL is degraded by the membrane-associated FtsH protease upon entry into the stationary phase, which will elicit LytA lysis by switching TA synthesis from LTA to WTA (25). Interestingly, the upregulation of TacL was accompanied by a 1.4-fold highly significant downregulation of FtsH in stationary cells with the *clpX_{I267E}* allele (Table S1), showing that the upregulation is growth-phase dependent. Therefore, the accumulation of TacL may be associated with the downregulation of FtsH. Alternatively, ClpXP may directly contribute to controlling TacL levels upon entry into stationary phase. Future studies will be directed to find direct substrates of the ClpXP protease.

## MATERIALS AND METHODS

### Cultivation and transformation

In this study, the serotype 2, *Streptococcus pneumoniae* strain D39 (CP027540.1) (83) with an *rpsL*$^+$ modification (+) was used. Wild-type and mutant derivatives were routinely grown in C + Y media (or Todd-Hewitt broth supplemented with yeast extract [THY]), or plated on 2% sheep blood Colombia agar plates at 37°C with 5% $CO_2$. Mutants were created utilizing the Sweet Janus cassette as described in reference 58. Linear DNA was synthesized using Phusion High-Fidelity Polymerase (New England Biolabs) and added to pre-competent cells of *S. pneumoniae* with 0.2 mg/mL competence-stimulating peptide

1. Transformants were plated on selective agar plates using either 250 µg/mL kanamycin for integration of the Sweet Janus cassette or 200 µg/mL streptomycin for removing the antibiotic resistance cassette. Primers used to generate mutants are listed in Table 1. Primer combinations are listed in Table 2, while overlap PCR products (used for cloning) are listed in Table 3.

## Growth curves and luminescence assays

Strains were grown until mid-exponential phase in C + Y media and subsequently diluted into a 96-well plate for a final $OD_{600}$ ~0.01 in a volume of 200 µL. For luminescence assay, C + Y was pH adjusted with 1 M HCl and was supplemented with 0.5 mg/mL D-luciferine potassium salt (Synchem, bc219). Temperature was set as required, and absorbance and luminescence were measured every 10 min.

## Simultaneous extended time-lapse microscopy

To assess the growth of multiple strains in real time, simultaneously, we employed a modified setup of time-lapse microscopy. A large 1.7 × 2.8 cm Gene Frame (Thermo Fisher Scientific, Massachusetts, USA) was attached to a clean microscope slide where a two-way septal split was created within the Gene Frame by attaching a fragmented Gene Frame, creating a four-well Gene Frame. C + Y with 1% agarose melted in an 84°C heating block, where melted C + Y agarose was transferred to the modified Gene Frame construction and immediately leveled by placing a clean microscope slide on top of the melted agarose mix. After 3–5 min (until the C + Y agarose solidified), the microscope slide was carefully removed. Using a clean scalpel, excess solidified C + Y agarose was removed from the septal areas and slide. Cultures of D39V $rpsL^+$ wild type and its mutant derivatives, $clpX_{I267E}$ and $ΔclpX$ $czcD::clpX$, were grown in C + Y at 37°C until an $OD_{600}$ of ~0.05 was obtained and were subsequently spotted onto each C + Y agarose pad. The spot was dried under a flame and sealed with a

**TABLE 1** Primer list for creation of mutants[a]

| Primer ID | Sequence (5′–3′) |
|---|---|
| 1kb_up_clpX_F | TGTTATCTACCGTTACTTGTCAGAAC |
| 1kb_down_clpX_R | CTTATATCAAAAGAAGTTTGGCAACAC |
| clpX_IEsub_F | GCCTTATTGTTTTGACCAAATCCctcGACTTTTTCACCCAGACGTTG |
| clpX_IEsub_R | GTCTGGGTGAAAAAGTCgagGGATTTGGTCAAAACAATAAG |
| clpX_janus_F | ggattatggccaatgaagactttactgCTCTAGGCTGTTTCTAGGATCG |
| clpX_janus_R | gtagtatttatgagagatggagcgaagGTGACTATGGAACTTAACACACAC |
| clpX_inf_R | ggatcggtttatccgttccatcGCCACAAAATGAGCAATAAACCATC |
| clpX_inf_F | GATGGAACGGATAAACCGATCC |
| clpX_rbs_czcD_R | cggagttatgagcatcaacattagaaaAATGGAAGGAAATCATGTCTACAAATAG |
| czcD_1kb_F | CTGTATAATCAGAAGGTTGACCAGG |
| czcD_janus_R | gtagtatttatgagagatggagcgaagCTTGGGTACTATCTTATTTGGAATAGAG |
| czcD_start_R | CAGGTGGAGTATTTGGTTCTAGC |
| czcD_end_F | TTTCTAATGTTGATGCTCATAACTCCG |
| czcD_end_clpX_R | cgatcctagaaacagcctagagCTTGGGTACTATCTTATTTGGAATAGAG |
| clpX_end_czcD_F | ctctattccaaataagatagtacccaagCTCTAGGCTGTTTCTAGGATCG |
| clpX_seqIGF_R | GTTTATCCGTTCCATCGACAG |
| lytA_1kb_F | CCTTCCTAATGACATGTCTGAATTG |
| lytA_1kb_R | GACCGCTATTTCTGCATCAATGAC |
| lytA_inf_R | GACAGGCCAGAATTCACAGTAG |
| lytA_inf_F | ctactgtgaattctggcctgtcCACGCCGACTTGAGGCAAATCa |
| lytA_jan_F | gtagtatttatgagagatggagcgaagCACGCCGACTTGAGGCAAATC |
| lytA_jan_R | ggattatggccaatgaagactttactgGACAGGCCAGAATTCACAGTAGAG |
| Janus_F | CAGTAAAGTCTTCATTGGCCATAATCC |
| Janus_R | CTTCGCTCCATCTCTCATAAATACTAC |

[a]Lower case letters indicate overhangs and/or non-annealing regions.

**TABLE 2** Primer combinations used for creating PCR products, later used in the creation of overlap constructs for transformations

| Forward primer | Reverse primer | Product |
|---|---|---|
| 1kb_up_clpX_F | clpX_IEsub_R | #1 A 1,824 bp product containing a 1 kb upstream region and an I–E mutation at position 267 in *clpX* |
| clpX_IEsub_F | 1kb_down_clpX_R | #2 A 1,481 bp product containing a 1 kb downstream region and an I–E mutation at position 267 in *clpX* |
| clpX_janus_F | clpX_IEsub_R | #3 A 454 bp product of the end of *clpX* while creating an I–E substitution at position 267 in *clpX*. An overhang on clpX_jan_F ensures homology to the Janus cassette. |
| 1kb_down_clpX_F | clpX_janus_R | #4 A 1,029 bp product homologous region downstream of *clpX*. An overhang on clpX_jan_R ensures homology to the Janus cassette. |
| clpX_janus_F | clpX_rbs_czcD_R | #5 A 1,249 bp product of *clpX* with its native RBS. An overhang on clpX_jan_F ensures homology to the Janus cassette. An overhang clpX_rbs_czcD_R ensures homology to *czcD* and czcD_end_F products. |
| czcD_1 kb_F | czcD_janus_R | #6 A 1,002 bp product of 1 kb downstream of czcD for homologous recombination. An overhang on czcD_janus_R ensures homology to the Janus cassette. |
| czcD_end_F | czcD_start_R | #7 A 824 bp product of *czcD* for homologous recombination. |
| czcD_1 kb_F | czcD_end_clpX_R | #8 A 1,002 bp product 1 kb downstream of czcD for homologous recombination. An overhang on czcD_end_clpX_R ensures homology of *czcD* to *clpX*. This overhang matches with clpX_end_czcD_F products. |
| clpX_end_czcD_F | clpX_rbs_czcD_R | #9 A 1,304 bp product of *clpX* with *czcD* overhangs in both ends. An overhang on clpX_end_czcD_F ensures homology to the *czcD* and czcD_end_clpX_R products. An overhang on clpX_rbs_czcD_R ensures homology to czcD_end_F products. |
| clpX_1 kb_up_F | clpX_inf_R | #10 A 1,051 bp product upstream of clpX, retaining the first 17 aa's of *clpX*. An overhang on the clpX_inf_R ensures overlap with clpX_inf_F. |
| clpX_inf_F | clpX_1 kb_down_R | #11 A 1,065 bp product including the last 11 aa's and a downstream region of *clpX*. |
| Janus_F | Janus_R | #12 A 1,444 bp product of the Janus cassette containing the *kanR* and *rpsL* genes |
| lytA_1 kb_F | lytA_jan_R | #13 A 780 bp product of the downstream region of *lytA*, retaining the last 16 aa's of *lytA*. An overhang on lytA_jan_R ensures homology to the Janus cassette. |
| lytA_jan_F | lytA_1 kb_R | #14 A 929 bp product of the upstream region of *lytA*, retaining the first 17 aa's of *lytA*. An overhang on lytA_jan_F ensures homology to the Janus cassette. |
| lytA_1 kb_F | lytA_inf_R | #15 A 773 bp product of the downstream region of *lytA,* retaining the last 16 aa's of *lytA* |
| lytA_inf_F | lytA_1 kb_R | #16 A 928 bp product of the upstream region of *lytA*, retaining the first 17 aa's of *lytA*. An overhang on lytA_inf_F ensures homology to lytA_inf_R. |

coverslip. An Olympus IX83 inverted microscope (Olympus LS) with a cellSens incubator module was used for temperature-specific growth. Here, a photometrics prime sCMOS camera and a 1.4 N.A. 100× oil-immersion objective were used for phase-contrast image acquisition. Images were acquired with a 20 ms exposure time every 5 min for 48–76 h at 37°C using the in-built time-lapse loop of the cellSens (Olympus LS) software. The microscope was programmed to move to pre-saved coordinates for each sample (three per strain) before imaging. In short, the microscope turns on the lamp, moves to a sample location, performs autofocus twice, captures a phase-contrast image, and then moves to a different pre-saved coordinate. The first autofocus step moves through an entire range of 40 µm in the Z-direction with 2 µm coarse steps and 1 µm fine steps. The second step includes fine-tuning through a range of 15 µm with 1 µm coarse steps of 0.5 µm fine-tuning steps. To avoid sample heating, shutters were closed between each image iteration. Videos were created using Fiji software (http://fiji.sc). Here, three locations were imaged for each strain at either pH (one replicate shown, per strain per condition). Full time-lapse videos were uploaded to Figshare: WT grown at 37°C in C + Y at pH 7.8 (https://doi.org/10.6084/m9.figshare.25562718.v1), $clpX_{I267E}$ grown on C + Y at pH 7.8 (https://doi.org/10.6084/m9.figshare.25562787.v1), and Δ*clpX czcD::clpX* grown on C + Y at pH 7.8 (https://doi.org/10.6084/m9.figshare.25562481.v1), and WT grown on C + Y at pH 7.0 (https://doi.org/10.6084/m9.figshare.25562868.v1), $clpX_{I267E}$ grown on C + Y at pH 7.0 (https://doi.org/10.6084/m9.figshare.25562889.v1), and Δ*clpX czcD::clpX* grown on C + Y at pH 7.0 (https://doi.org/10.6084/m9.figshare.25562907.v1).

**TABLE 3** Overlap PCR products using products in Table 2[a]

| Overlap PCRs | Primers | Product |
|---|---|---|
| #1 and #2 | 1kb_up_clpX_F, 1kb_down_clpx_R | A 3,256 bp markerless region of *clpX* with an I267E substitution; used for the creation of *clpX$_{I267E}$* |
| #1, #3, #4, and #12, | 1kb_up_clpX_F, 1kb_down_clpx_R | A 4,700 bp region of *clpX* with an I267E substitution and a Janus cassette attached in between *clpX* and *engB*; used for the creation of *clpX$_{I267E}$ kanR* |
| #6, #12, #5, and #7 | czcD_1kb_F, czcD_start_R | A 4,519 bp region of *czcD::clpX* with a Janus cassette at the end of *clpX*; used for the creation of *czcD::clpX kanR* |
| #8, #9, and #7 | czcD_1kb_F, czcD_start_R | A 3,075 bp region of *czcD::clpX* without a Janus cassette at the end of *clpX*. Used for creation of *czcD::clpX* |
| #10 and #11 | 1kb_up_clpX_F, 1kb_down_clpx_R | A 2,110 bp region, a 1 kb upstream and downstream region of *clpX*, with a *clpX* in-frame deletion retaining the first 17 and last 11 aa's of *clpX*. This product was later used for attaching Janus overhangs using clpX_janus_F, 1kb_up_clpX_R, and 1kb_down_clpX_F, clpX_janus_R, overlapping these products with #12; used for the creation of Δ*clpX kanR czcD::clpX* and later Δ*clpX czcD::clpX* |
| #13, #12, and #14 | lytA_1kb_F, lytA_1kb_R | A 3,145 bp region containing the up- and downstream regions of *lytA* with a Janus cassette inserted into the *lytA* gene; used for the creation of Δ*lytA kanR* |
| #15 and #16 | lytA_1kb_F, lytA_1kb_R | A 1,702 bp region of a markerless in-frame deletion of *lytA*, retaining the first 17 and last 16 aa's; used for the creation of Δ*lytA* |

[a]The primer combination and a description of the resulting product are indicated in the columns.

## *Galleria mellonella* infection model

The infection of *Galleria mellonella* (Reptilienkosmos, Germany) was conducted as described recently (27). Briefly, D39V *rpsL$^+$* wild type and its mutant derivatives, *clpX$_{I267E}$*, Δ*clpX czcD::clpX*, and Δ*clpX czcD::clpX* +200 µM ZnCl$_2$, were grown in THY medium until mid-exponential phase (OD$_{600}$ = 0.35–0.45) and washed with 0.9% sodium chloride. Then 10 µL of the bacterial suspension containing $3.0 \times 10^5$ bacterial cells was used to infect the larvae having a weight of 0.3–0.4 g via the intrahemocelic route. A gastight microliter syringe (Hamilton, USA), coupled with a repeating dispenser (Hamilton), was used to ensure equal infection doses. Each group of 10 larvae was incubated at 37°C with sufficient food for 7 days post-infection and monitored daily. Food contained wheat bran, oatmeal, dry yeast, skim milk powder, honey, and glycerol in variable proportions.

## Immunoblotting

Proteins were extracted either from the cellular fraction or supernatant of a pneumococcal culture. For cellular protein fractions, pellets from 40 mL culture volumes were normalized to their exact OD$_{600}$ values and French pressed twice. For extracellular proteins, 25 mL of the supernatant was precipitated overnight in 1:1 vol of cold 96% ethanol. Protein pellet was resuspended in 50 mM Tris-HCl, 100 mM NaCl, and 0.5% SDS and normalized to the sample's exact OD$_{600}$ value. Proteins were separated on 4%–12% Bis-Tris pre-cast NuPAGE gels (Thermo Fisher Scientific) in MOPS SDS-PAGE buffer. Proteins were transferred to a polyvinylidene difluoride (PVDF) membrane in transfer buffer with methanol (10 mM Tris-HCl, 100 mM glycine, and 10% methanol) for 1 h (300 mA, 20 V). Membranes were blocked in 5% skim milk and phosphate-buffered saline (PBS)-Tween for 1 h. Membranes were washed briefly with PBS-Tween before the addition of primary antibody, using either ClpX polyclonal mouse antibody (1:4,000), polyclonal LytA rabbit antibody (1:10,000), rabbit anti-pneumolysin (1:5,000), or rabbit anti-enolase (1:5,000) in 2% bovine serum albumin (BSA) and PBS-Tween. PVDF membranes were incubated overnight with primary antibody at 4°C. Membranes were subsequently washed 3× with PBS-Tween. Appropriate horseradish peroxidase (HRP)-conjugated secondary antibody (Dako) (1:6,000) in PBS-Tween was applied for 1 h before washing 3×. Then, Immobilon Forte HRP substrate was added and immediately imaged using Amersham ImageQuant 800 GxP.

## Teichoic acid labeling and image analysis

Strains of WT $rpsL^+$, $clpX_{I267E}$, and $\Delta clpX$ $czcD::clpX$ (with or without zinc) were grown in 4 mL C + Y media until an $OD_{600}$ of ~0.02–0.05 was obtained. Next, 5 µg/mL 1-azi-doethyl-choline (Jenna Bioscience) and 20 µM sDIBO Alexa 488 (Sigma-Aldrich) were added and grown at 37°C for 1–2 h (for exponential phase samples, OD ~0.3–0.4) or left in stationary phase for 2 h before imaging. Cells were spotted onto a leveled C + Y + 1% agarose pad and imaged using an Olympus IX83 inverted microscope with a 1.4 N.A. ×100 oil immersion objective. Phase-contrast and flourescence (for the sDIBO-488) images were acquired in succession and processed in Fiji (v.216/1.54p).

Images of teichoic acid labeled strains were processed using the MicrobeJ (v.5.13 p) plugin for Fiji (v.2.16/1.54p) (ImageJ) (84). Cells were detected using the following settings: area (0.5–4.0 µm$^2$), length (0.5–4.0 µm), width (0.5–3.0 µm), with segmentation, shape descriptors, exclude on edges, straighten, shape, profile, and type enabled. Under the type tab, four extra criteria were made to ensure grouping based on cell length. Here, cells were segmented into criteria 1: [SHAPE.length] ≤1.277, criteria 2: [SHAPE.length] >1.277 and [SHAPE.length] ≤1.577, criteria 3: [SHAPE.length] >1.577 and [SHAPE.length] ≤2.245, and criteria 4: [SHAPE.length] >2.245 as described in reference 85. Detected cells were subject to manual editing to ensure cells were correctly segmented before loading into the results tab. Here, sufficient biological replicate images were loaded to achieve $n$ ~1,000 cells. For the heatmaps: Using the Shape Plot function, categories were set for Type.index and Data: Straighten.ch2 (channel containing the fluorescent images). Fluorescent intensities were normalized, and relative intensities were set to range from 0.75 to 1.00 in the properties tab. An offset of 250p was set to avoid overlapping graphs. For the demographs: Using the demograph plot function, data were set to Straighten.ch2 and sorted according to SHAPE.length. The fluorescent signal was normalized, and the range was set from 0.4 to 1.0 in the properties tab.

## RNA purification and qPCR

Strains were grown in C + Y at pH 7.8 until indicated $OD_{600}$ values were obtained at 37°C. Cultures were briefly submerged into liquid nitrogen before being pelleted at 4°C. RNA was purified from snap-frozen pellets using hot phenol-chloroform RNA extraction as described in reference 86. Briefly, cells were resuspended in 150 µL solution 1 (10 mM Na-acetate, 10 mM Na-citrate, and 2 mM EDTA) and transferred to a tube of 700 µL phenol (pH 4.5), 300 µL chloroform, and 150 µL solution 2 (10 mM Na-acetate, pH 4.5, and 2% SDS). Tubes were inverted four to six times and heated at 80°C for 5 min. Tubes were briefly cooled and spun at 10,000 × $g$ for 5 min at room temperature (RT). The aqueous phase was transferred to a tube of 500 µL chloroform and vortexed for 20 seconds. Tubes were spun at 10,000 × $g$ for 5 min, and the aqueous phase was precipitated overnight in 96% ethanol. RNA was pelleted for 45 min in a pre-cooled centrifuge, washed twice in 70% ethanol, and resuspended in nuclease-free $H_2O$. Total RNA of 1,000 ng was treated with DNase I (New England Biolabs) and subsequently heat inactivated. RNA was reverse transcribed to cDNA using reverse transcriptase (Applied Biosystems) according to manufacturer's instructions. RT-qPCR was performed using $lytA$- and $16S$-specific primers, in technical triplicate using 5 µL RealQ Plus 2× Master Mix Green (Ampliqon), 0.5 µL of each primer (10 µM primer stock), and 4 µL diluted cDNA in a 384-well plate. The plate was sealed with transparent film and run using LightCycler 480. Expression of $lytA$ was subsequently normalized to 16S and quantified compared to the WT at an $OD_{600}$ of ~0.2.

## TEM

*S. pneumoniae* strains were pre-grown in C + Y at pH 7.0 until mid-log phase, then diluted in 5 mL C + Y at pH 7.8 to a starting $OD_{600}$ of ~0.05. Cultures were grown until an $OD_{600}$ of ~0.4 and pelleted for 5 min at 5,000 × $g$. Cells were washed in PBS, and the

pellet was resuspended in fixation solution (2.5% glutaraldehyde in 0.1 M cacodylate buffer, pH 7.4) and incubated overnight at 4°C. Cells were further treated with 2% osmium tetroxide and 0.25% uranyl acetate for contrast enhancement. The pellets were subsequently dehydrated using stepwise increasing concentrations of ethanol, followed by the addition of pure propylene oxide, and embedded in Epon resin. Thin sections for electron microscopy were stained with lead citrate and observed in a Philips CM100 BioTWIN transmission electron microscope fitted with an Olympus Veleta camera with a resolution of 2,048 by 2,048 pixels. Sample processing and microscopy were performed at the Core Facility for Integrated Microscopy, Faculty of Health and Medical Sciences, University of Copenhagen.

## SEM of pneumococcal strains

The pneumococcal strains, D39V $rpsL^+$ wild type, and its mutant derivatives, $clpX_{I267E}$ and $\Delta clpX$ $czcD::clpX$, were cultivated in RPMI-based minimal medium [RPMI1640 with 30 mM glucose, 21 mM $NaHCO_3$, 1 mM glycine, 240 µM choline chloride, 1.7 mM $NaH_2PO_4$, 3.8 mM $Na_2HPO_4$, 2 mM adenine, 5 mM uracil, 300 µM $MnSO_4$, 165 µM $FeSO_4$, and 22 µM $Fe(NO_3)_3$] with the addition of 0.25% yeast extract. The cultivation was performed in pre-warmed minimal medium with an inoculum of optical density at $OD_{600}$ 0.08. Cells of all strains were harvested by centrifugation at $2,000 \times g$ at $OD_{600}$ 0.4, washed with PBS, and fixed with 2.5% glutaraldehyde and 2% paraformaldehyde in washing buffer (0.1 M cacodylate buffer [pH 7], 0.09 M sucrose, 0.01 M $CaCl_2$, and 0.01 M $MgCl_2$) for 1 h at 4°C. After that, cells were attached onto poly-L-lysine-coated coverslips for 90 min, and then fixation continued at 4°C overnight. Samples were washed with washing buffer three times for 5 min each time, treated with 1% osmium tetroxide in washing buffer for 1 h at RT, and then washed again with washing buffer three times for 5 min each time. Afterward, the samples were dehydrated in a graded series of aqueous ethanol solutions (10%, 30%, 50%, 70%, 90%, and 100%) on ice for 15 min for each step. Before the final change to 100% ethanol, the samples were allowed to reach RT, and then critical point drying with liquid $CO_2$ was performed. Finally, the samples were mounted on aluminium stubs, sputtered with gold/palladium, and examined with a field emission scanning electron microscope, Supra 40VP (Carl Zeiss Microscopy Deutschland GmbH, Oberkochen, Germany), using the Everhart-Thornley SE detector and the inlens detector at an 80:20 ratio at an acceleration voltage of 5 kV. All micrographs were edited by using Adobe Photoshop CS6.

## Sample harvesting and preparation for mass-spectrometric analysis

The proteomic data set was generated and analyzed as part of an ongoing study aiming at identifying substrates targeted for degradation by ClpP by each of the four different Clp ATPases in pneumococcus. The proteomic data in Fig. 6 compare changes in the whole cell proteome between the D39V WT and D39V $clpX_{I267E}$. Both strains harbor the wild-type $clpP$ gene in the chromosome. Additionally, both strains carry the same gene insertion, harboring an additional mutant variant of the $clpP$ gene encoding for a proteolytically inactive ClpP-TRAP that cannot interact with WT ClpP because it has two exchanges of amino acid residues at the interface used for ClpP-ClpP interaction (87). D39V WT and D39V $clpX_{I267E}$ harboring a constitutively expressed ClpP-TRAP (ClpP with the following modifications: S96A, H142E, E117R, and a C-terminal 6× His-tag) construct inserted at the chromosomal expression platform (88) locus were pre-grown in modified RPMI medium [RPMI1640 and 30 mM glucose, 21 mM $NaHCO_3$, 1 mM glycine, 240 µM choline chloride, 1.7 mM $NaH_2PO_4$, 3.8 mM $Na_2HPO_4$, 2 mM adenine, 5 mM uracil, 300 µM $MnSO_4$, 165 µM $FeSO_4$, 22 µM $Fe(NO_3)_3$] at 37°C in a water bath and diluted to a starting $OD_{600}$ of 0.2 in 40 mL pre-heated modified RPMI. Fifteen OD units ($OD_{600}$ × volume = $OD_{600}$ unit) were harvested at both exponential and stationary growth phases. Exponential phase samples were harvested at an $OD_{600}$ of ~0.5 to 0.6, and stationary-phase samples were collected at an $OD_{600}$ of ~1.5, all in biological triplicate.

Cells were pelleted in a pre-cooled centrifuge and washed twice with 10 mL PBS. Pellets were kept frozen in a −80°C freezer until cell disruption.

For cell disruption, the pellet was suspended in 20 mM HEPES, pH 8.0, and added to a frozen vinyl bead-beating chamber containing liquid nitrogen. A clean metal ball was added, and the chamber sealed. Cells were beaten for 2 min at 3.0 m/s in a bead mill (Retsch GmbH, Germany). The powdered cells were resuspended in 20 mM HEPES at pH 8.0. The lysates were treated with Pierce Universal Nuclease (Pierce, Thermo Fisher Scientific; 2.5 U, 4 mM $MgCl_2$) for 25 min at 37°C. Cell debris was removed by centrifugation (30 min, 17,000 × $g$, RT).

## Preparation for mass spectrometry

Protein concentration of cellular fractions was determined using the Micro BCA Protein Assay Kit (Pierce, Thermo Fisher Scientific) according to reference 89.

Following the digestion of proteins and purification of peptides, mass spectrometry (MS) was performed according to references 89 and 90 with minor adaptations. In brief, 5 µg of the cellular protein fraction was incubated with 100 µg of hydrophilic (GE Healthcare, Little Chalfont, UK) and hydrophobic (Thermo Fisher Scientific) carboxylate-modified magnetic SeraMag Speed Beads and shaken in 80% (vol/vol) acetonitrile (ACN) for 10 min at room temperature. The beads were then washed twice with 80% (vol/vol) ethanol and once with 100% ACN. For protein digestion, the beads were rebuffered into 50 mM Tris-HCl 1 mM $CaCl_2$ (pH 8.0) and incubated with 200 ng trypsin/LysC Mix (Promega, Madison, USA) for 16 h at 37°C. The digestion was stopped, and peptides were eluted by the addition of 0.5% trifluoroacetic acid.

## Mass spectrometric measurements and data analysis

For liquid chromatography-tandem mass spectrometry (LC-MS/MS) analyses, the tryptic peptide solutions were separated on an Ultimate 3000 RSLC system (Thermo Fisher Scientific) and analyzed in data-independent acquisition (DIA) mode on an Orbitrap Exploris 480 (Thermo Fisher Scientific). For further details, see Table S2A and B.

DIA MS data were searched against the *S. pneumoniae* D39V protein database (fasta file generated from National Center for Biotechnology Information RefSeq NZ_CP027540 annotation) comprising 1,861 pneumococcal proteins and the contaminant proteins trypsin and benzonase, as well as the marker proteins ClpP_TRAP, KanR, and RpsL_2, using the Spectronaut software (v.18.6.231227.55695; Biognosys AG, Schlieren, Switzerland) with settings described in Table S2C in the directDIA analysis type, in accordance with the procedure described by reference 91. The mass spectrometry proteomics data have been deposited to the ProteomeXchange Consortium via the PRIDE (92) partner repository with the data set identifier PXD061622.

Global median-normalized Spectronaut-processed data were further analyzed in R (v.4.1.2) using an in-house developing version of the SpectroPipeR package (93). Proteins identified with at least two peptides were considered. The peptide-based ROPECA (94) statistics were calculated protein-wise for the mutant compared to the WT strain. Protein levels were considered to differ significantly between strains if the fdr-adjusted $P$ value ($Q$ value) was less than 0.05 and the absolute fold change was at least 2. Functional annotation and naming of the proteins were obtained from pneumowiki.med.uni-greifswald.de.

## ACKNOWLEDGMENTS

We are very thankful to the Bernhardt-Rudner supergroup (Department of Microbiology, Harvard Medical School) for the generous gift of the LytA antibodies and to Morten Kjos (Norwegian University of Life Sciences) for providing the *ssbB-luc* PCR product. We also thank the staff at the Core Facility for Integrated Microscopy (University of Copenhagen) for their assistance in imaging.

## AUTHOR AFFILIATIONS

[1]Department of Veterinary and Animal Sciences, University of Copenhagen, Copenhagen, Capital Region of Denmark, Denmark

[2]Research Unit of Molecular Microbiology, University of Southern Denmark, Odense, Region Syddanmark, Denmark

[3]Department of Molecular Genetics and Infection Biology, Interfaculty Institute for Genetics and Functional Genomics, Center for Functional Genomics of Microbes, University of Greifswald, Greifswald, Mecklenburg-Vorpommern, Germany

[4]Department of Functional Genomics, Interfaculty Institute for Genetics and Functional Genomics, Center for Functional Genomics of Microbes, University Medicine Greifswald, Greifswald, Mecklenburg-Vorpommern, Germany

[5]Imaging Center of the Department of Biology, University of Greifswald, Greifswald, Mecklenburg-Vorpommern, Germany

## AUTHOR ORCIDs

Viktor H. Mebus  http://orcid.org/0009-0009-2332-2866
Uwe Völker  http://orcid.org/0000-0002-5689-3448
Dorte Frees  http://orcid.org/0000-0003-4946-2890

## FUNDING

| Funder | Grant(s) | Author(s) |
| --- | --- | --- |
| Independent research fund Denmark, Technology and Production | 0136-00200 | Dorte Frees |
| Independent Research Fund Denmark Tecknology and Production | 4264-00028B | Dorte Frees |
| Deutsche Forschungsgemeinschaft | DFG-GRK 2719/1 | Uwe Völker |
| | | Sven Hammerschmidt |

## AUTHOR CONTRIBUTIONS

Viktor H. Mebus, Conceptualization, Data curation, Formal analysis, Methodology, Visualization, Writing – original draft | Supradipta De, Data curation, Formal analysis, Methodology, Writing – original draft | Larissa M. Busch, Data curation, Formal analysis, Methodology, Writing – original draft | Manuela Gesell Salazar, Data curation, Methodology | Rabea Schlüter, Methodology, Supervision | Uwe Völker, Conceptualization, Formal analysis, Funding acquisition, Supervision, Writing – original draft | Sven Hammerschmidt, Conceptualization, Methodology, Supervision, Writing – original draft | Dorte Frees, Conceptualization, Formal analysis, Funding acquisition, Project administration, Supervision, Visualization, Writing – original draft

## ADDITIONAL FILES

The following material is available online.

### Supplemental Material

**Supplemental figures (Spectrum00804-25-s0001.pdf).** Fig. S1 to S4.
**Table S1 (Spectrum00804-25-s0002.xlsx).** Proteomic data.
**Table S2 (Spectrum00804-25-s0003.pdf).** Settings used in microscopy.

### Open Peer Review

**PEER REVIEW HISTORY (review-history.pdf).** An accounting of the reviewer comments and feedback.

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
