## [Reviewer comments · Microbiology Spectrum]

Microbiology Spectrum

The ClpXP protease and the ClpX unfoldase control virulence, cell division and autolysis in *Streptococcus pneumoniae*

Viktor Mebus, Supradipta De, Larissa Busch, Manuela Gesell Salazar, Rabea Schlüter, Uwe Voelker, Sven Hammserschmidt, and Dorte Frees

Corresponding Author(s): Dorte Frees, Kobenhavns Universitet

Review Timeline:

Submission Date:	March 19, 2025
Editorial Decision:	April 7, 2025
Revision Received:	April 11, 2025
Accepted:	April 21, 2025

Editor: Carlos Blondel

Reviewer(s): The reviewers have opted to remain anonymous.

Transaction Report:

DOI: <https://doi.org/10.1128/spectrum.00804-25>

Re: Spectrum00804-25 (**The ClpXP protease and the ClpX unfoldase control virulence, cell division and autolysis in *Streptococcus pneumoniae***)

Dear Prof. Dorte Frees:

Thank you for the privilege of reviewing your work. Below you will find my comments, instructions from the Spectrum editorial office, and the reviewer comments.

Revision Guidelines

Sincerely,
Carlos Blondel
Editor
Microbiology Spectrum

Reviewer #1 (Comments for the Author):

In their very interesting manuscript "The ClpXP protease and the ClpX unfoldase control virulence, cell division and autolysis in *Streptococcus pneumoniae*" Mebus et al investigate the different phenotypes and possible functions of the ClpXP AAA+ protease complex for this Gram+ pathogen.

ClpX consists of an N-terminal domain followed by an AAA+ domain, which forms a hexamer, binds ATP and uses ATP

hydrolysis to unfold substrate proteins, which can be directly targeted via degrons or by adaptor proteins, which can recognize substrate and interact e.g. via the NTD to target them to ClpX unfolding. Furthermore, to form a ClpXP AAA+ protease complex, the hexameric ClpX unfoldase can associate with ClpP double heptameric protease complex via peptide loops extending from the bottom of ClpX, interacting via the IGF tip with its binding sites on the top surface of the ClpP complex. During this association the unfolded egressing substrate polypeptide chain can be directly transported into the center of the ClpP complex lined by active peptidase sites to hydrolyze the incoming unfolded polypeptide chains.

ClpX is encoded by an essential gene and seem to have a pleiotropic impact on *Streptococcus pneumoniae* physiology, therefore the authors used two very specific elegant mutant strains for their comprehensive investigation on the cellular function of clpX.

By changing the Isoleucine of the IGF binding motif to EGF the ClpX1267E variant interferes with the association of the ClpX peptide loop with ClpP. Thereby the protease activity of ClpX-ClpP is abolished, but both proteins are otherwise kept functional. ClpX kept its substrate recognition, ATPase and unfoldase activity and the full functional ClpP is still able to interact with its other cognate AAA+ unfoldases such as ClpC.

In an additional strain the expression of clpX was put under the control of a Zn-dependent promoter at an ectopic place in trans allowing the deletion of the original clpX gene. With this Δ clpX czcD::clpX strain they can investigate the role of ClpX by depleting the cellular amount of ClpX via controlling the Zn²⁺ dependent promoter.

In summary, the Δ clpX czcD::clpX strain is to a certain extent comparable to a Δ clpX strain, and the strain the ClpX loop variant I267E is devoid of the specific ClpXP protease activity. (Fig 1).

I have some comments, remarks or questions

-Intro p2 I63 Maybe it would help to better understand the possible ClpP independent function of ClpX by briefly mentioning this early example of an ClpX (unfoldase) mediated disassembly of a phage transposase complex (Levchenko et al (1995) *Genes Dev* 9:2399). Or maybe mention it later in the discussion on this (p11 I329-334), where this mechanism would give a more concrete and clear example of a process directly supported only by the unfoldase activity.

-Results p 5 "Depletion of ClpX is selective for mutations in the SPV_1595 gene" Please mention already here and not only much later (p10) in the discussion that "the SPV_1595 gene that together with the neighboring SPV_1594 gene has been proposed to encode a horizontally acquired toxin/antitoxin addiction module"

-p5 I126; p9 I249-50; p11 I329, I342 include protease (activity): "inhibiting ClpXP" protease "activity" and p8 I241 "inactivation of ClpXP" protease activity is associated...

-p5 I147 "previous studies indicate that ClpXP suppresses competence development" Is it in the absence or presence of ClpXP?

-p6 I 163-165 "In agreement with this notion, lysis was abrogated when the *lytA* gene was deleted showing that *LytA* is causing the accelerated lysis of cells devoid of ClpX or ClpXP"

Is this just the presence or the activation or lack of inhibition/degradation of *LytA*?

-p6 I208-210 "the accelerated autolysis observed for D39V devoid of ClpXP activity correlates with greatly enhanced levels of extracellular *LytA*, and that the upregulation of *LytA* takes place at the post-transcriptional level and is not linked to increased cell lysis."

Can the authors elaborate a little on what could be meant by "post-transcriptional" (activation/secretion/stability?)

-p7 I200 "(Fig 3A, Exponential cultures: clpX1267E: P-value = 0.058, Δ clpX czcD::clpX: P-value = 0.224, stationary phase: clpX1267E: P-value = 0.054, Δ clpX czcD::clpX: P-value = 0.308)" Why is that written here? and not in the figure legend?

-p8 I225-6 I would delete: ("The two step one-pot labeling method" described by"

-p10 I281-289 Is it known whether in *S. pneumoniae* suppressor mutants in *spx* (suppressor of clpP and clpX) are appearing in clpX or clpP mutant strains? You already mention that one *SpxA2* (but not *SpxA1*) might get degraded by ClpXP. Or is it only the mentioned SPV_1595 gene that gets mutated?

-I285 abundance not "abundacne"

-I289-91 I would suggest to delete this somehow confusing sentence: "However, because of the phenotypes described above, we focused our attention on differentially abundant proteins with a confirmed role in cell division, TA biosynthesis, and virulence - see discussion." The discussion starts right after this sentence...

Discussion p10 I308 Maybe "our experiments suggest /indicate / demonstrate" instead of "we indicate that ClpXP"

Reviewer #2 (Comments for the Author):

In the manuscript by Mebus et al., the authors have explored a chaperone protease complex called ClpXP involved in protein homeostasis in a wide range of organisms. ClpP multimers form the proteolytic center and ClpX is an ATP dependent chaperon that unfolds proteins and feeds them into the ClpP protease. Here the authors have studied ClpXP from the human pathogen *Streptococcus pneumoniae*. The authors used mutants in which a ClpXP complex cannot form and a clpX depletion mutant to show that ClpX and ClpXP are important for several functions in pneumococci, i.e virulence, cell division and controlling the levels of extracellular LytA (the major pneumococcal autolysin). In addition, click chemistry was used to show changed pattern of choline on the teichoic acids of ClpXP mutants. Finally, a proteomics approach was used to identify proteins with altered levels in ClpXP cells. Among others, proteins involved in teichoic acid synthesis and other cell division proteins are highlighted. The manuscript presents novel results on which cell functions ClpXP affects in pneumococci and that ClpX also has a ClpP-independent function. The experiments are robust, and the results are presented in a very well-written manuscript with informative figures. Nevertheless, I have some comments that I hope the authors can address.

1. The clpX depletion strain acquired a suppressor mutation resulting in a truncated SPV_1595 protein. Is the depletion strain the parental strain of the ClpX(I67E) mutant? Based on line 138-139, I assume this is the case; that both strains have the suppressor. This is an important point in order to rule out the possibility that differences observed between the depletion mutant and ClpX(I67E) are not a result from this suppressor only being present in the depletion strain. Did the authors also try to make a single Δ SPV_1595 mutant and check that it e.g. does not grow in chains or displays premature autolysis?
2. In many of the experiments a Zn(2+)-induced control is not included (except Fig. 2A and Fig. S3). It would have been nice to show that induction of the ectopic clpX expression restores a normal phenotype. For example, a violin plot in Fig. 4B showing that cell size distribution is restored (or at least partly restored) upon Zn(2+)-induction.
3. It is a very interesting observation that the extracellular levels of LytA increases dramatically in ClpXP cells. The intracellular levels seem unchanged between the WT and the mutants, which suggests that the total production (and release) of LytA increases in the ClpXP cells. However, transcription is not increased. How can this be explained? Increased translation rates or other possibilities?
4. Was the Zn(2+)-induced cells washed prior to injection to the *G. mellonella*? Will 200 μ M ZnCl₂ have an effect the larvae?
5. The choline moieties on TA were elegantly visualized by using click-chemistry. In the manuscript the authors describe this as the pattern of TA, however, the pattern of choline on TA can be modified by a phosphorylcholine esterase called Pce. Did the author see any changes in the level of Pce in their proteomic analyses that could influence the choline patterns observed? If not, this should be mentioned in the text.

Line117: pneumococcus should not be in italic.

Line132: Change "Zn" to "Zn(2+)".

Line134: Change "cells expressing" to "cells ectopically expressing".

Line188: Should be "virulence".

Line199-201: It was not clear from what data the p-values were calculated.

Line430: Some of this text should not be in italic.

Line431: "A600" should be "OD600".

Line444: Change "run" to "separated".

Line499: Remove line break.

Line521: change rpm to how many g forces or include what type of centrifuge.

Line536-538: Did the authors use the ClpP-Trap version of ClpP for the proteomics analyses? Based on the Fig. 6 and text in the results section, it was not clear that this was a pull-down of ClpP-trapped protein targets.

Line575: ClpP-TRAP?

We would first like to thank the editor and two reviewers for the fast review proces. We also greatly appreciate the reviewer's enthusiasm for the work and the insightful comments that have helped us to improve the manuscript. Please find below a detailed point-by-point answer to each comment in *grey shaded* (line numbers refer to changes in the marked- up copy version of the revised manuscript).

Reviewer #1 (Comments for the Author):

In their very interesting manuscript "The ClpXP protease and the ClpX unfoldase control virulence, cell division and autolysis in *Streptococcus pneumoniae*" Mebus et al investigate the different phenotypes and possible functions of the ClpXP AAA+ protease complex for this Gram+ pathogen. ClpX consists of an N-terminal domain followed by an AAA+ domain, which forms a hexamer, binds ATP and uses ATP hydrolysis to unfold substrate proteins, which can be directly targeted via degrons or by adaptor proteins, which can recognize substrate and interact e.g. via the NTD to target them to ClpX unfolding.

Furthermore, to form a ClpXP AAA+ protease complex, the hexameric ClpX unfoldase can associate with ClpP double heptameric protease complex via peptide loops extending from the bottom of ClpX, interacting via the IGF tip with its binding sites on the top surface of the ClpP complex. During this association the unfolded egressing substrate polypeptide chain can be directly transported into the center of the ClpP complex lined by active peptidase sites to hydrolyze the incoming unfolded polypeptide chains.

ClpX is encoded by an essential gene and seem to have a pleiotropic impact on *Streptococcus pneumoniae* physiology, therefore the authors used two very specific elegant mutant strains for their comprehensive investigation on the cellular function of clpX.

By changing the Isoleucine of the IGF binding motif to EGF the ClpXI267E variant interferes with the association of the ClpX peptide loop with ClpP. Thereby the protease activity of ClpX-ClpP is abolished, but both proteins are otherwise kept functional. ClpX kept its substrate recognition, ATPase and unfoldase activity and the full functional ClpP is still able to interact with its other cognate AAA+ unfoldases such as ClpC.

In an additional strain the expression of clpX was put under the control of a Zn-dependent promoter at an ectopic place in trans allowing the deletion of the original clpX gene. With this Δ clpX czcD::clpX strain they can investigate the role of ClpX by depleting the cellular amount of ClpX via controlling the Zn²⁺ dependent promoter.

In summary, the Δ clpX czcD::clpX strain is to a certain extent comparable to a Δ clpX strain, and the strain the ClpX loop variant I267E is devoid of the specific ClpXP protease activity. (Fig 1).

I have some comments, remarks or questions

-Intro p2 l63 Maybe it would help to better understand the possible ClpP independent function of ClpX by briefly mentioning this early example of an ClpX (unfoldase) mediated disassembly of a phage transposase complex (Levchenko et al (1995) Genes Dev 9:2399). Or maybe mention it later in the discussion on this (p11 l329-334), where this mechanism would give a more concrete and clear example of a process directly supported only by the unfoldase activity.

Author response: this is an excellent suggestion and we now refer to the MuA transposase study in line 65-67.

-Results p 5 "Depletion of ClpX is selective for mutations in the SPV_1595 gene" Please mention already here and not only much later (p10) in the discussion that "the SPV_1595 gene that together with the neighboring SPV_1594 gene has been proposed to encode a horizontally acquired toxin/antitoxin addiction module"

Author response: agreed, we have moved these line to the introduction, line 140-42.

-p5 l126; p9 l249-50; p11 l329, l342 include protease (activity): "inhibiting ClpXP" protease "activity" and p8 l241 "inactivation of ClpXP" protease activity is associated...

Author response: corrected.

-p5 l147 "previous studies indicate that ClpXP suppresses competence development" Is it in the absence or presence of ClpXP?

Author response: we have now revised these lines to clarify that "ClpXP contributes to inhibition of competence development under non-permissive conditions" (line 151-153).

-p6 l 163-165 "In agreement with this notion, lysis was abrogated when the *lytA* gene was deleted showing that *LytA* is causing the accelerated lysis of cells devoid of ClpX or ClpXP"
Is this just the presence or the activation or lack of inhibition/degradation of *LytA*?

Author response: for now, we do not know the exact mechanism but we are in the process of testing if *LytA* is a direct substrate of ClpXP.

-p6 l208-210 "the accelerated autolysis observed for D39V devoid of ClpXP activity correlates with greatly enhanced levels of extracellular *LytA*, and that the upregulation of *LytA* takes place at the post-transcriptional level and is not linked to increased cell lysis."

Can the authors elaborate a little on what could be meant by "post-transcriptional" (activation/secretion/stability?)

Author response: for clarification, these lines have been rephrased: "We conclude that the accelerated autolysis observed for D39V devoid of ClpXP activity correlates with greatly enhanced levels of extracellular *LytA* despite that transcription of the *lytA* gene and cellular levels of *LytA* are not affected in cells lacking ClpXP proteolytic activity. For now, the mechanisms underlying the increase in extracellular *LytA* levels remain obscure, however, we have ruled out that it is caused by increased cell lysis", line 211-215.

-p7 l200 "(Fig 3A, Exponential cultures: *clpX*l267E: P-value = 0.058, Δ *clpX* *czcD*::*clpX*: P-value = 0.224, stationary phase: *clpX*l267E: P-value = 0.054, Δ *clpX* *czcD*::*clpX*: P-value = 0.308)" Why is that written here? and not in the figure legend?

Author response: this sentence has been moved to the legend for Fig. 3A as suggested.

-p8 l225-6 I would delete: ("The two step one-pot labeling method" described by"

Author response: corrected

-p10 l281-289 Is it known whether in *S. pneumoniae* suppressor mutants in *spx* (suppressor of *clpP* and *clpX*) are appearing in *clpX* or *clpP* mutant strains? You already mention that one *SpxA2* (but not *SpxA1*) might get degraded by ClpXP. Or is it only the mentioned SPV_1595 gene that gets mutated?

Author response: mutations in the *spxA2* gene were not observed in our *clpX* depletion strain or the strain lacking ClpXP proteolytic activity.

-l285 abundance not "abundacne"

Author response: corrected

-l289-91 I would suggest to delete this somehow confusing sentence: "However, because of the phenotypes described above, we focused our attention on differentially abundant proteins with a confirmed role in cell division, TA biosynthesis, and virulence - see discussion." The discussion starts right after this sentence...

Author response: corrected

Discussion p10 l308 Maybe "our experiments suggest /indicate / demonstrate" instead of "we indicate that ClpXP"

Author response: has been corrected to: "Here, we suggest that ClpXP is contributing to virulence as pneumococcus carrying a ClpX variant that cannot associate with ClpP displays reduced lethality in the *G. melonella* larvae model of infection, line 308-310.

Reviewer #2 (Comments for the Author):

In the manuscript by Mebus et al., the authors have explored a chaperone protease complex called ClpXP involved in protein homeostasis in a wide range of organisms. ClpP multimers form the proteolytic center and ClpX is an ATP dependent chaperon that unfolds proteins and feeds them into the ClpP protease. Here the authors have studied ClpXP from the human pathogen *Streptococcus pneumoniae*. The authors used mutants in which a ClpXP complex cannot form and a *clpX* depletion mutant to show that ClpX and ClpXP are important for several functions in pneumococci, i.e virulence, cell division and controlling the levels of extracellular LytA (the major pneumococcal autolysin). In addition, click chemistry was used to show changed pattern of choline on the teichoic acids of ClpXP mutants. Finally, a proteomics approach was used to identify proteins with altered levels in ClpXP cells. Among others, proteins involved in teichoic acid synthesis and other cell division proteins are highlighted. The manuscript presents novel results on which cell functions ClpXP affects in pneumococci and that ClpX also has a ClpP-independent function. The experiments are robust, and the results are presented in a very well-written manuscript with informative figures. Nevertheless, I have some comments that I hope the authors can address.

1. The *clpX* depletion strain acquired a suppressor mutation resulting in a truncated SPV_1595 protein. Is the depletion strain the parental strain of the ClpX(I67E) mutant? Based on line 138-139, I assume this is the case; that both strains have the suppressor. This is an important point in order to

rule out the possibility that differences observed between the depletion mutant and ClpX(I67E) are not a result from this suppressor only being present in the depletion strain. Did the authors also try to make a single Δ SPV_1595 mutant and check that it e.g. does not grow in chains or displays premature autolysis?

Author response: For clarification, we have rephrased this paragraph line 130-144 to more clearly describe that the ClpX(I67E) mutant was constructed in the D39V wt background and that the strain only deviates from the wt by the desired genetic changes in the *clpX* gene. In contrast, WGS of the *clpX* depletion strain showed that the strain additionally has acquired a SNP in SPV_1595. Hence, phenotypes associated with ClpX depletion are deduced from comparing D39V Δ *clpX* *czcD::clpX* grown in the absence of Zn(2+) (ClpX depleted) or presence of Zn(2+) (ClpX ectopically expressed) which allows us to rule out a contribution from the truncation of SPV_1595.

2. In many of the experiments a Zn(2+)-induced control is not included (except Fig. 2A and Fig. S3). It would have been nice to show that induction of the ectopic *clpX* expression restores a normal phenotype. For example, a violin plot in Fig. 4B showing that cell size distribution is restored (or at least partly restored) upon Zn(2+)-induction.

Author response: please find the violin plot showing that ectopic expression of ClpX restores normal cell size distribution in the revised Fig. 4B. We updated the text accordingly (line 239-242).

3. It is a very interesting observation that the extracellular levels of LytA increase dramatically in ClpXP cells. The intracellular levels seem unchanged between the WT and the mutants, which suggests that the total production (and release) of LytA increases in the ClpXP cells. However, transcription is not increased. How can this be explained? Increased translation rates or other possibilities?

Author response: we have clarified that the underlying mechanism remains unknown, line 211-215: “We conclude that the accelerated autolysis observed for D39V devoid of ClpXP activity correlates with greatly enhanced levels of extracellular LytA despite that transcription of the *lytA* gene and cellular levels of LytA are not affected in cells lacking ClpXP proteolytic activity. For now, the mechanisms underlying the increase in extracellular LytA levels remain obscure, however, we have ruled out that it is caused by increased cell lysis”, line 211-215”.

4. Was the Zn(2+)-induced cells washed prior to injection to the *G. mellonella*? Will 200 μ M ZnCl₂ have an effect the larvae?

Author response: ZnCl₂ was added only to the THY medium during cultivation of the D39 Δ *clpX* complementation strain. Following cultivation, the bacteria were washed once with 0.9% NaCl, and the infection dose was adjusted to 5×10^5 CFU in 0.9% NaCl. Therefore, no Zn²⁺ was present in the final infection dose. For this reason, 200 μ M ZnCl₂ alone was not tested for toxicity in larvae. The control group was injected with 0.9% NaCl alone.

5. The choline moieties on TA were elegantly visualized by using click-chemistry. In the manuscript the authors describe this as the pattern of TA, however, the pattern of choline on TA can be modified by a phosphorylcholine esterase called Pce. Did the author see any changes in the level of Pce in their proteomic analyses that could influence the choline patterns observed? If not, this should be mentioned in the text.

Author response: . The proteomics dataset shows a non-significant increase in Cpe= CbpE levels in the D39V *clpX_{I267E}* strain in both growth phases (2.1 folds in exponential samples/2.9 in stationary phase; q-value ~ 0.9/0.5). This information has now been included in the main text, line 364-366.

Line117: pneumococcus should not be in italic. **Author response:** corrected

Line132: Change "Zn" to "Zn(2+)". **Author response:** corrected, line 145 in the revised manuscript.

Line134: Change "cells expressing" to "cells ectopically expressing". **Author response:** corrected

Line188: Should be "virulence". **Author response:** corrected

Line199-201: It was not clear from what data the p-values were calculated. **Author response: this is now clarified in lines,**

Line430: Some of this text should not be in italic. **Author response: corrected**

Line431: "A600" should be "OD600". **Author response: corrected**

Line444: Change "run" to "separated". **Author response: corrected**

Line499: Remove line break. **Author response: corrected**

Line521: change rpm to how many g forces or include what type of centrifuge. **Author response: corrected**

Line536-538: Did the authors use the ClpP-Trap version of ClpP for the proteomics analyses? Based on the Fig. 6 and text in the results section, it was not clear that this was a pull-down of ClpP-trapped protein targets.

Author response: as stated in line 578-579, the proteomic data set was generated and analyzed as part of an ongoing study aiming at identifying substrates targeted for degradation by ClpP by each of the four different Clp ATPases in pneumococcus. The proteomic data in Fig. 6 compare changes in the whole cell proteome between the D39V WT and D39V *clpX_{I265E}*. Both strains harbor the wild-type *clpP* gene in the chromosome. Additionally, both strains an additional mutant variant of the *clpP* gene encoding for a proteolytically inactive ClpPTRAP that cannot interact with WT ClpP because it has two exchanges of amino acid residues at the interface used for ClpP-ClpP interaction. The strain was constructed with inspiration from Trentini et al. (2016*) and enable us to identify substrates associated with ectopically expressed ClpPTRAP in cells with a functional ClpP. This important information is now provided in line, 541-548.

Line575: ClpP-TRAP? **Author response:** explained above.

Re: Spectrum00804-25R1 (**The ClpXP protease and the ClpX unfoldase control virulence, cell division and autolysis in *Streptococcus pneumoniae***)

Dear Prof. Dorte Frees:

Your manuscript has been accepted, and I am forwarding it to the ASM production staff for publication. Your paper will first be checked to make sure all elements meet the technical requirements. ASM staff will contact you if anything needs to be revised before copyediting and production can begin. Otherwise, you will be notified when your proofs are ready to be viewed.

Sincerely,
Carlos Blondel
Editor
Microbiology Spectrum